# Solver-in-the-Loop: Learning from Differentiable Physics to Interact with Iterative PDE-Solvers

**Kiwon Um**[1,2]    **Robert Brand**[1]    **Yun (Raymond) Fei**[3]    **Philipp Holl**[1]    **Nils Thuerey**[1]

[1]Technical University of Munich, [2]LTCI, Telecom Paris, IP Paris, [3]Columbia University

kiwon.um@telecom-paris.fr, robert.brand@tum.de
yf2320@columbia.edu, {philipp.holl, nils.thuerey}@tum.de

## Abstract

Finding accurate solutions to partial differential equations (PDEs) is a crucial task in all scientific and engineering disciplines. It has recently been shown that machine learning methods can improve the solution accuracy by correcting for effects not captured by the discretized PDE. We target the problem of reducing numerical errors of iterative PDE solvers and compare different learning approaches for finding complex correction functions. We find that previously used learning approaches are significantly outperformed by methods that integrate the solver into the training loop and thereby allow the model to interact with the PDE during training. This provides the model with realistic input distributions that take previous corrections into account, yielding improvements in accuracy with stable rollouts of several hundred recurrent evaluation steps and surpassing even tailored supervised variants. We highlight the performance of the differentiable physics networks for a wide variety of PDEs, from non-linear advection-diffusion systems to three-dimensional Navier-Stokes flows.

## 1 Introduction

Numerical methods are prevalent in science to improve the understanding of our world, with applications ranging from climate modeling [55, 53] over simulating the efficiency of airplane wings [47] to analyzing blood flow in a human body [27]. These applications are extremely costly to compute due

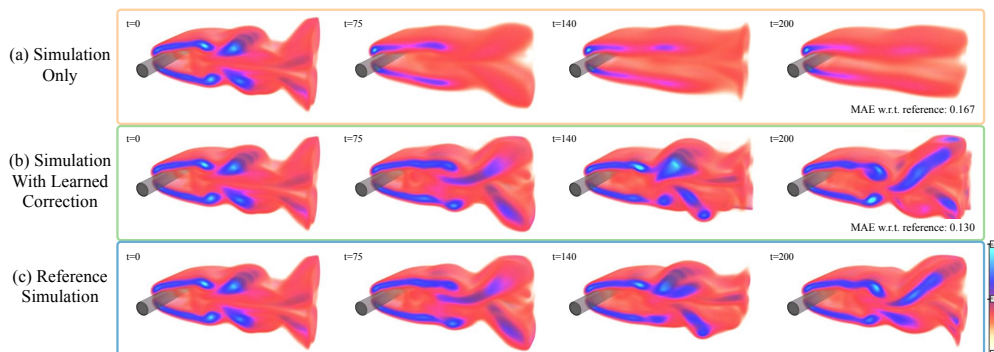

Figure 1: A 3D fluid problem (shown in terms of vorticity) for which the regular simulation introduces numerical errors that deteriorate the resolved dynamics (a). Combining the same solver with a learned corrector trained via differentiable physics (b) significantly reduces errors w.r.t. the reference (c).

to the fine spatial and temporal resolutions required in real-world scenarios. In this context, deep learning methods are receiving strongly growing attention [40, 4, 18] and show promise to account for those components of the solutions that are difficult to resolve or are not well captured by our physical models. Physical models typically come in the form of PDEs and are discretized in order to be processed by computers. This step inevitably introduces numerical errors. Despite a vast amount of work [15, 2] and experimental evaluations [7, 41], analytic descriptions of these errors remain elusive for most real-world applications of simulations.

In our work, we specifically target the numerical errors that arise in the discretization of PDEs. We show that, despite the lack of closed-form descriptions, discretization errors can be seen as functions with regular and repeating structures and, thus, can be learned by neural networks. Once trained, such a network can be evaluated locally to improve the solution of a PDE-solver, i.e., to reduce its numerical error.

The core of most numerical methods contains some form of iterative process – either in the form of repeated updates over time for explicit solvers or even within a single update step for implicit solvers. Hence, we focus on iterative PDE solving algorithms [17]. We show that neural networks can achieve excellent performance if they take the reaction of the solver into account. This interaction is not possible with supervised learning on pre-computed data alone. Even small inference errors of a supervised model can quickly accumulate over time [57, 29], leading to a data distribution that differs from the distribution of the pre-computed data. For supervised learning methods, this causes deteriorated inference at best and solver explosions at worst.

We demonstrate that neural networks can be successfully trained if they can *interact* with the respective PDE solver during training. To achieve this, we leverage differentiable simulations [1, 58]. Differentiable simulations allow a trained model to autonomously explore and experience the physical environment and receive directed feedback regarding its interactions throughout the solver iterations. Hence, our work fits into the broader context of machine learning as differentiable programming, and we specifically target recurrent interactions of highly non-linear PDEs with deep neural networks. This combination bears particular promise: it improves generalizing capabilities of the trained models by letting the PDE-solver handle large-scale changes to the data distribution such that the learned model can focus on localized structures not captured by the discretization. While physical models generalize very well, learned models often specialize in data distributions seen at training time. However, we will show that, by combining PDE-based solvers with a learned model, we can arrive at hybrid methods that yield improved accuracy while handling solution manifolds with significant amounts of varying physical behavior.

We show the advantages of training via differentiable physics for explicit and implicit solvers applied to a broad class of canonical PDEs. For explicit and semi-implicit solvers, we consider advection-diffusion systems as well as different types of Navier-Stokes variants. We showcase models trained with up to 128 steps of a differentiable simulator and apply our model to complex three-dimensional (3D) flows, as shown in Fig. 1. Additionally, we present a detailed empirical study of different approaches for training neural networks in conjunction with iterative PDE-solvers for recurrent rollouts of several hundred time steps. On the side of implicit solvers, we consider the Poisson problem [37], which is an essential component of many PDE models. Here, our method outperforms existing techniques on predicting initial guesses for a conjugate gradient (CG) solver by receiving feedback from the solver at training time. The source code for this project is available at `https://github.com/tum-pbs/Solver-in-the-Loop`.

**Previous Work**   Combining machine learning techniques with PDE models has a long history in machine learning [13, 28, 8]. More recently, deep-learning-based methods were successfully applied to infer stencils of advection-diffusion problems [4], to discover PDE formulations [35, 42, 52], and to analyze families of Poisson equations [36]. While identifying governing equations represents an interesting and challenging task, we instead focus on a general method to improve the solutions of chosen spaces of solutions.

Other studies have investigated the similarities of dynamical systems and deep learning methods [65] and employed conservation laws to learn systems described by Hamiltonian mechanics [18, 12]. Existing studies have also identified discontinuities in finite-difference solutions with deep learning [46] and focused on improving the iterative behavior of linear solvers [24]. So-called Koopman operators likewise represent an interesting opportunity for deep learning algorithms [40, 32]. While these methods replace the PDE-based time integration with a learned version, our models rely on and

interact with a PDE-solver that provides a coarse approximation to the problem. Hence, our models always alternate between inference via an artificial neural network (ANN) and a solver step. This distinguishes our work from studies of recurrent ANN architectures [11, 54, 62] as the PDE-solver can introduce significant non-linearities in-between evaluations of the ANN.

We focus on chaotic systems for which fluid flow represents an exciting and challenging problem domain that is highly relevant for industrial applications. Deep learning methods have received significant amounts of attention in this area [31]. For example, both steady [19] and unsteady [40], as well as multi-phase flows [16] have been investigated with deep learning based approaches. Turbulence closure modeling has been an area of particular focus [59, 38, 6]. Additionally, convolutional neural networks (CNNs) were studied for stochastic sub-grid modeling [60], airfoil flow problems [56, 67], and as part of generative networks to leverage the fast inference of pre-trained models [10, 66, 29]. Other studies have targeted the unsupervised learning of divergence-free corrections [57] or incorporated PDE-based loss functions to represent individual flow solutions via ANNs [43, 52]. In addition to temporal predictions of turbulent flows [39], similar algorithms were more recently also employed for classification problems [20]. However, to the best of our knowledge, the existing methods do not let ANNs interact with solver in a recurrent manner. As we will demonstrate below, this combination yields significant improvements in terms of inference accuracy.

While we focus on Eulerian, i.e., grid-based discretizations, the Lagrangian viewpoint is a popular alternative. While a variety of studies has investigated graph-based simulators, e.g., for rigid-body physics in the context of human reasoning [5, 64, 3] or weather predictions [51], particles are also a popular basis for fluid flow problems [33, 61, 48]. Despite our Eulerian focus, Lagrangian methods could likewise benefit from incorporating differentiable solvers into the training process.

Our work shares the motivation of previous work to use differentiable components at training time [1, 14, 58, 9] and frameworks for differentiable programming [50, 25, 26, 23]. Differentiable physics solvers were proposed for inverse problems in the context of liquids [49], cloth [34], soft robots [25], and molecular dynamics [63]. While these studies typically focus on optimization problems or replace solvers with learned components, we focus on the interaction between the two. Hence, in contrast to previous work, we always rely on a PDE-solver to yield a coarse approximate solution and improve its performance via a trained ANN.

## 2    Learning to Reduce Numerical Errors

Numerical methods yield approximations of a smooth function $\boldsymbol{u}$ in a discrete setting and invariably introduce errors. These errors can be measured in terms of the deviation from the exact analytical solution. For discrete simulations of PDEs, they are typically expressed as a function of the truncation, $O(\Delta t^k)$. Higher-order methods, with large $k$, are preferable but difficult to arrive at in practice. For practical schemes, no closed-form expression exists for truncation errors, and the errors often grow exponentially as solutions are integrated over time. We investigate methods that solve a discretized PDE $\mathcal{P}$ by performing discrete time steps $\Delta t$. Each subsequent step can depend on any number of previous steps, $\boldsymbol{u}(\boldsymbol{x}, t + \Delta t) = \mathcal{P}(\boldsymbol{u}(\boldsymbol{x}, t), \boldsymbol{u}(\boldsymbol{x}, t - \Delta t), ...)$, where $\boldsymbol{x} \in \Omega \subseteq \mathbb{R}^d$ for the domain $\Omega$ in $d$ dimensions, and $t \in \mathbb{R}^+$.

**Problem Statement:**  We consider two different discrete versions of the same PDE $\mathcal{P}$, with $\mathcal{P}_R$ denoting a more accurate discretization with solutions $\mathbf{r} \in \mathscr{R}$ from the *reference manifold*, and an approximate version $\mathcal{P}_s$ with solutions $\boldsymbol{s} \in \mathscr{S}$ from the *source manifold*. We consider $\mathbf{r}$ and $\boldsymbol{s}$ to be states at a certain instance in time, i.e., they represent phase space points, and evolutions over time are given by a trajectory in each solution manifold. As we focus on the discrete setting, a solution over time consists of a *reference sequence* $\{\mathbf{r}_t, \mathbf{r}_{t+\Delta t}, \cdots, \mathbf{r}_{t+k\Delta t}\}$ in the solution manifold $\mathscr{R}$, and correspondingly, a more coarsely approximated *source sequence* $\{\boldsymbol{s}_t, \boldsymbol{s}_{t+\Delta t}, \cdots, \boldsymbol{s}_{t+k\Delta t}\}$ exists in the solution manifold $\mathscr{S}$. We also employ a mapping operator $\mathcal{T}$ that transforms a phase space point from one solution manifold to a suitable point in the other manifold, e.g., for the initial conditions of the sequences above, we typically choose $\boldsymbol{s}_t = \mathcal{T}\mathbf{r}_t$. We discuss the choice of $\mathcal{T}$ in more detail in the appendix, but in the simplest case, it can be obtained via filtering and re-sampling operations.

By evaluating $\mathcal{P}_R$ for $\mathscr{R}$, we can compute the points of the phase space sequences, e.g., $\mathbf{r}_{t+\Delta t} = \mathcal{P}_R(\mathbf{r}_t)$ for an update scheme that only depends on time $t$. Without loss of generality, we assume a fixed $\Delta t$ and denote a state $\mathbf{r}_{t+k\Delta t}$ after $k$ steps of size $\Delta t$ with $\mathbf{r}_{t+k}$. Due to the inherently different numerical approximations, $\mathcal{P}_s(\mathcal{T}\mathbf{r}_t) \neq \mathcal{T}\mathbf{r}_{t+1}$ for the vast majority of states. In chaotic systems, such differences typically grow exponentially over time until they saturate at the level

of mean difference between solutions in the two manifolds. We use an $L^2$-norm in the following to quantify the deviations, i.e., $\mathcal{L}(\boldsymbol{s}_t, \mathcal{T}\mathbf{r}_t) = \|\boldsymbol{s}_t - \mathcal{T}\mathbf{r}_t\|_2$. Our learning goal is to arrive at a correction operator $\mathcal{C}(\boldsymbol{s})$ such that a solution to which the correction is applied has a lower error than an unmodified solution: $\mathcal{L}(\mathcal{P}_s(\mathcal{C}(\mathcal{T}\mathbf{r}_{t_0})), \mathcal{T}\mathbf{r}_{t_1}) < \mathcal{L}(\mathcal{P}_s(\mathcal{T}\mathbf{r}_{t_0}), \mathcal{T}\mathbf{r}_{t_1})$. The correction function $\mathcal{C}(\boldsymbol{s}|\theta)$ is represented as a deep neural network with weights $\theta$ and receives the state $\boldsymbol{s}$ to infer an additive correction field with the same dimension. To distinguish the original phase states $\boldsymbol{s}$ from corrected ones, we denote the latter with $\tilde{\boldsymbol{s}}$, and we use an exponential notation to indicate a recursive application of a function, i.e.,

$$\boldsymbol{s}_{t+n} = \mathcal{P}_s(\mathcal{P}_s(\cdots \mathcal{P}_s(\mathcal{T}\mathbf{r}_t)\cdots)) = \mathcal{P}_s^n(\mathcal{T}\mathbf{r}_t) . \tag{1}$$

Within this setting, any type of learning method naturally needs to compare states from the source domain with the reference domain in order to bridge the gap between the two solution manifolds. How the evolution in the source manifold at training time is computed, i.e., if and how the corrector interacts with the PDE, has a profound impact on the learning process and the achievable final accuracy. We distinguish three cases: no interaction, a pre-computed form of interaction, and a tight coupling via a differentiable solver in the training loop.

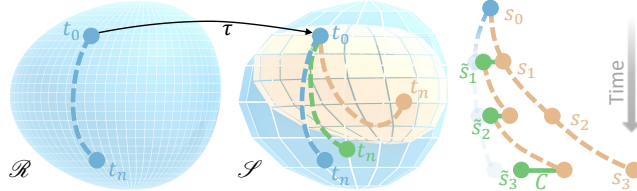

Figure 2: Transformed solutions of the reference sequence computed on $\mathscr{R}$ (blue) differ from solutions computed on the source manifold $\mathscr{S}$ (orange). A correction function $\mathcal{C}$ (green) updates the state after each iteration to more closely match the projected reference trajectory on $\mathscr{S}$.

- **Non-interacting (NON)**: The learning task purely uses the unaltered PDE trajectories, i.e., $\boldsymbol{s}_{t+n} = \mathcal{P}_s^n(\mathcal{T}\mathbf{r}_t)$ with $n$ evaluations of $\mathcal{P}_s$. These trajectories are fully contained in the source manifold $\mathscr{S}$. Learning from these states means that a model will not see any states that deviate from the original solutions. As a consequence, models trained in this way can exhibit undesirably strong error accumulations over time. This corresponds to learning from the difference between the orange and blue trajectories in Fig. 2, and most commonly applied supervised approaches use this variant.

- **Pre-computed interaction (PRE)**: To let an algorithm learn from states that are closer to those targeted by the correction, i.e., the reference states, a pre-computed or analytic correction is applied. Hence, the training process can make use of phase space states that deviate from those in $\mathscr{S}$, as shown in green in Fig. 2, to improve inference accuracy and stability. This approach can be formulated as $\boldsymbol{s}_{t+n} = (\mathcal{P}_s \mathcal{C}_{\text{pre}})^n(\mathcal{T}\mathbf{r}_t)$ with a pre-computed correction function $\mathcal{C}_{\text{pre}}$. In this setting, the states $\boldsymbol{s}$ are corrected without employing a neural network, but they should ideally resemble the states achievable via the learned correction later on. As the modified states $\boldsymbol{s}$ are not influenced by the learning process, the training data can be pre-computed. A correction model $\mathcal{C}(\boldsymbol{s}|\theta)$ is trained via $\tilde{\boldsymbol{s}}$ that replaces $\mathcal{C}_{\text{pre}}$ at inference time.

- **Solver-in-the-loop (SOL)**: By integrating the learned function into a differentiable physics pipeline, the corrections can interact with the physical system, alter the states, and receive gradients about the future performance of these modifications. The learned function $\mathcal{C}$ now depends on states that are modified and evolved through $\mathcal{P}$ for one or more iterations. A trajectory for $n$ evaluations of $\mathcal{P}_s$ is given by $\tilde{\boldsymbol{s}}_{t+n} = (\mathcal{P}_s\mathcal{C})^n(\mathcal{T}\mathbf{r}_t)$, with $\mathcal{C}(\tilde{\boldsymbol{s}}|\theta)$. The key difference with this approach is that $\mathcal{C}$ is trained via $\tilde{\boldsymbol{s}}$, i.e., states that were affected by previous evaluations of $\mathcal{C}$, and it affects $\tilde{\boldsymbol{s}}$ in the following iterations. As for (PRE), this learning setup results in a trajectory like the green one shown in Fig. 2, however, in contrast to before, the learned correction itself influences the evolution of the trajectory, preventing a gap for the data distribution of the inputs.

In addition to these three types of interaction, a second central parameter is the look-ahead trajectory per iteration and mini-batch of the optimizer during learning. A subscript $n$ denotes the number of steps over which the future evolution is recursively evaluated, e.g., $\text{SOL}_n$. The objective function, and hence the quality of the correction, is evaluated with the training goal to minimize $\sum_{i=t}^{t+n} \mathcal{L}(\boldsymbol{s}_i, \mathbf{r}_i)$. Below, we will analyze a variety of learning methodologies that are categorized via learning methodology (NON, PRE or SOL) and look-ahead horizon $n$.

# 3 Experiments

We now provide a summary and discussion of our experiments with the different types of PDE interactions for a selection of physical models. Full details of boundary conditions, parameters, and discretizations of all five PDE scenarios are given in App. B.

## 3.1 Model Equations and Data Generation

We investigate a diverse set of constrained advection-diffusion models of which the general form is

$$\partial \boldsymbol{u}/\partial t = -\boldsymbol{u} \cdot \nabla \boldsymbol{u} + \nu \nabla \cdot \nabla \boldsymbol{u} + \mathbf{g} \quad \text{subject to} \quad \boldsymbol{M}\boldsymbol{u} = 0, \tag{2}$$

where $\boldsymbol{u}$ is the velocity, $\nu$ denotes the diffusion coefficient (i.e., viscosity), and $\mathbf{g}$ denotes external forces. The constraint matrix $\boldsymbol{M}$ contains an additional set of equality constraints imposed on $\boldsymbol{u}$. In total, we target four scenarios: pure non-linear advection-diffusion (Burger's equation), two-dimensional Navier-Stokes flow, Navier-Stokes coupled with a second advection-diffusion equation for a buoyancy-driven flow, and a 3D Navier-Stokes case. Also, we discuss CG solvers in the context of differentiable operators below.

For each of the five scenarios, we implement the non-interacting evaluation (NON) by pre-computing a large-scale data set that captures a representative and non-trivial space of solutions in $\mathscr{S}$. The reference solutions from $\mathscr{R}$ are typically computed with the same numerical method using a finer discretization (4x in our setting, with effective resolutions of $128^2$ and higher). The PDEs are parametrized such that the change of discretization leads to substantial differences when integrated over time. For several of the 2D scenarios, we additionally train models with data sets of trajectories that have been corrected with other pre-computated correction functions. For these PRE variants, we use a time-regularized, constrained least-squares corrector [21] to obtain corrected phase state points. For the SOL variants, we employ a differentiable PDE-solver that runs mini-batches of simulations and provides gradients for all operations of the solving process within the deep learning framework. This allows gradients to freely propagate through the PDE-solver and coupled neural networks via automatic differentiation. For $n > 1$, i.e., PDE-based look-ahead at training time, the gradients are back-propagated through the solver $n - 1$ times, and the difference w.r.t. a pre-computed reference solution is evaluated for all intermediate results.

## 3.2 Training Procedure

The neural network component $F(\boldsymbol{s}\,|\,\theta)$ of the correction function is realized with a fully convolutional architecture. As our focus lies on the methodology for incorporating PDE models into the training, the architectures are intentionally kept simple. However, they were chosen to yield high accuracy across all variants. Our networks typically consist of 10 convolutional layers with 16 features each, interspresed with ReLU activation functions using kernel sizes of $3^d$ and $5^d$. The networks parameters $\theta$ are optimized with a fixed number of steps with an ADAM optimizer [30] and a learning rate of $10^{-4}$. For validation, we use data sets generated from the same parameter distribution as the training sets. All results presented in the following use test data sets whose parameter distributions differ from the ones of the training data set.

We quantify the performance of the trained models by computing the mean absolute error between a computed solution and the corresponding projected reference for $n$ consecutive steps of a simulation. We report absolute error values for different models in comparison to an unmodified source trajectory from $\mathscr{S}$. Additionally, relative improvements are given w.r.t. the difference between unmodified source and reference solutions. An improvement by 100% would mean that the projected reference is reproduced perfectly, while negative values indicate that the modified solution deviates more from the reference than the original source trajectory.

# 4 Results

Our experiments show that learned correction functions can achieve substantial gains in accuracy over a regular simulation. When training the correction functions with differentiable physics, this additionally yields further improvements of more than 70% over supervised and pre-computed approaches from previous work. A visual overview of the different tests is given in Fig. 3, and a summary of the full evaluation from the appendix is provided in Fig. 4 and Table 1. In the appendix, we also provide error measurements w.r.t. physical quantities such as kinetic energy and frequency content. The source code of our experiments and analysis will be published upon acceptance.

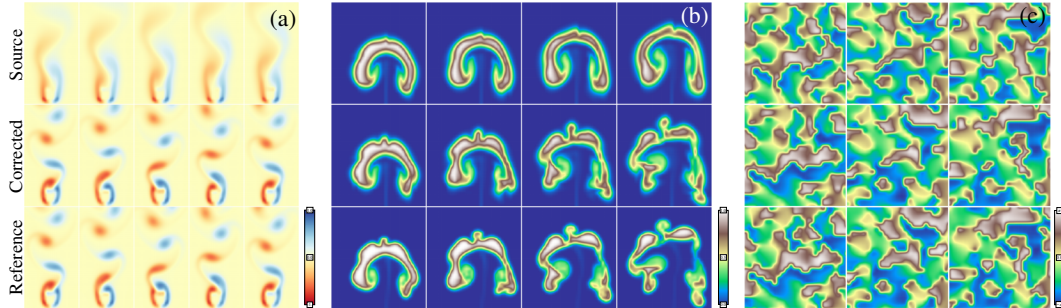

Figure 3: Our PDE scenarios cover a wide range of behavior including (a) vortex shedding, (b) complex buoyancy effects, and (c) advection-diffusion systems. Shown are different time steps (l.t.r.) in terms of vorticity for (a), transported density for (b), and angle of velocity direction for (c).

**Unsteady Wake Flow**     The PDE scenario for unsteady wake flows represents a standard benchmark case for fluids [44, 40] and involves a continuous inflow with a fixed, circular obstacle, which induces downstream vortex shedding with distinct frequencies depending on the Reynolds number. For coarse discretizations, the approximation errors distort the flow leading to deteriorated motions or suppressed vortex shedding altogether. An example flow configuration is shown in Fig. 3a. In this scenario, the simplest method (NON) yields stable training and a model that already reduces the mean absolute error (MAE) from 0.146 for a regular simulation without correction (SRC) to an MAE of 0.049 when applying the learned correction. The pre-computed correction (PRE) improves on this behavior via its time regularization with an error of 0.031. A $SOL_{32}$ model trained with a differentiable physics solver for 32 time steps in each iteration of ADAM yields a significantly lower error of 0.013. This means, the numerical errors of the source simulation w.r.t. the reference were reduced by more than a factor of 10. Despite the same architecture and weight count for all three models, the overall performance varies strongly, with the $SOL_{32}$ version outperforming the simpler variants by 73% and more. An example of the further evaluations provided in the appendix is given in Fig. 4h.

**Buoyancy-driven Flow**     We evaluate buoyancy-driven flows as a scenario with increased complexity. In addition to an incompressible fluid, a second, non-uniform marker quantity is advected with the flow that exerts a buoyancy force. This coupled system of equations leads to interesting and complex swirling behavior over time. We additionally use this setup to highlight that the reference solutions can be obtained with different discretization schemes. We use a higher-order advection scheme in addition to a $4\times$ finer spatial discretization to compute the reference data.

Interestingly, the correction functions benefit from particularly long rollouts at training time in this scenario. Models with simple pre-computed or unaltered trajectories yield mean errors of 1.37 and 1.07 compared to an error of 1.59 for the source simulation, respectively. Instead, a model trained with differentiable physics with 128 steps ($SOL_{128}$) successfully reduces the error to 0.62, an improvement of more than 59% compared to the unmodified simulation.

**Forced Advection-Diffusion**     A third scenario employs Burger's equation as a physical model. We mimic the setup from previous work [4] to inject energy into the system via a forcing term with a spectrum of sine waves. This forcing prevents the system from dissipating to relatively static and slowly moving configurations. While the PRE and NON versions yield clear improvements, the SOL versions do not significantly outperform the simpler baselines. This illustrates a limitation of long rollouts via differentiable physics: Learned correction functions need to be able to anticipate future behavior to make high-quality corrections. The randomized forcing in this example severely limits the number of future steps that can accurately be predicted given one state. This behavior contrasts with other physical systems without external disturbances, where a single state uniquely determines its evolution. We show in the appendix that the SOL models with an increased number of interaction steps pay off when the external disturbances are absent.

**Conjugate Gradient Solver**     We turn to iterative solvers for linear systems of equations to illustrate another aspect of learning from differentiable physics: its importance for the propagation of boundary condition effects. As our learning objective, we target the inference of initial guesses for CG solvers [22]. Following previous work [57], we target Poisson problems of the form $\nabla \cdot \nabla p = \nabla \cdot \boldsymbol{u}$, which arise for projections of a velocity $\boldsymbol{u}$ to a divergence-free state. Instead of fully relying on an ANN

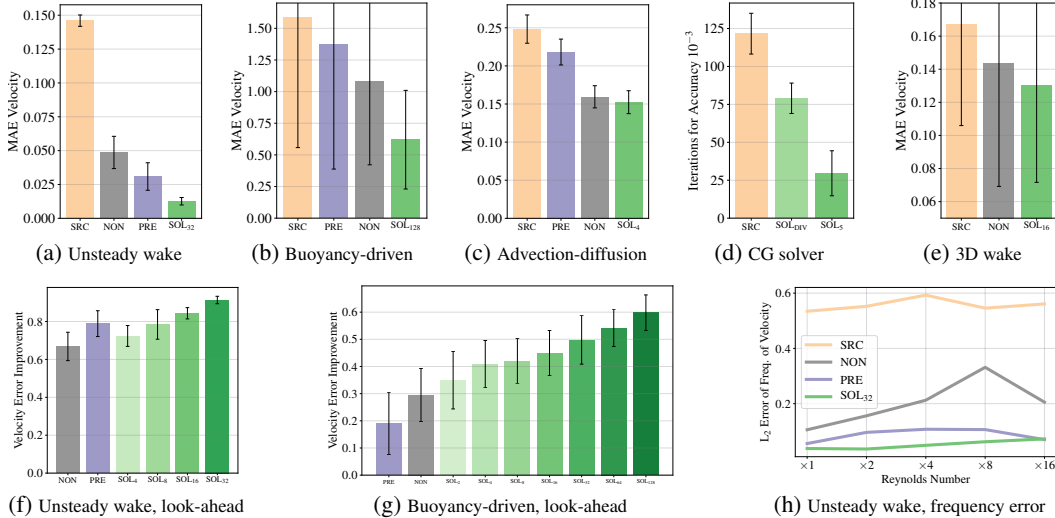

Figure 4: (a)-(e) Numerical approximation error w.r.t. reference solution for unaltered simulations (SRC) and with learned corrections. The models trained with differentiable physics and look-ahead achieve significant gains over the other models. (f,g) Relative improvement over varying look-ahead horizons. (h) A frequency-based evaluation for the unsteady wake flow scenario.

to produce the pressure field $p$, we instead target the learning objective to produce an initial guess, which is improved by a regular CG solver until a given accuracy threshold is reached.

This goal can be reached by directly minimizing the right-hand side term $\nabla \cdot \boldsymbol{u}$, similar to physics-based loss terms proposed in a variety of studies [43, 52]. Alternatively, we can employ a differentiable CG solver and formulate the learning goal as minimizing the same residual after $n$ steps of the CG solver (similar to the $\text{SOL}_n$ models above). While the physics-based loss version reduces the initial divergence more successfully, it fares badly when interacting with the CG solver: compared to the SOL version, it requires 63% more steps to reach a desired accuracy. Inspecting the inferred solutions reveals that the former model leads to comparatively large errors near boundaries, which are small for each grid cell but significantly influence the solution on a large scale. The SOL version immediately receives feedback about this behavior via the differentiable solver iterations. I.e., the differentiable solver provides a look-ahead of how different parts of the solution affect future states. In this way, it can anticipate problems such as those in the vicinity of boundary conditions.

**Three-dimensional Fluid Flow**  Lastly, we investigate a 3D case of incompressible flow. The overall setup is similar to the unsteady wake flow in two dimensions outlined above, but the third dimension extends the axes of rotation in the fluid from one to three, yielding a very significant increase in complexity. As a result, the flow behind the cylindrical obstacle quickly becomes chaotic and forms partially turbulent eddies, as shown in Fig. 1. This scenario requires significantly larger models to learn a correction function, and the NON version does not manage to stabilize the flow consistently. Instead, the $\text{SOL}_{16}$ version achieves stable rollouts for several hundred time steps and successfully corrects the numerical inaccuracies of the coarse discretization, improving the numerical accuracy of the source (SRC) simulation by more than 22% across a wide range of configurations.

## 5   Ablations and Discussion

We performed an analysis of the proposed training via differentiable physics to highlight which hyperparameters most strongly influence results. Specifically, we evaluate varying look-ahead horizons, different model architectures, training via perturbations, and pre-computed variants.

**Future Look-Ahead**  For systems with deterministic behavior, long rollouts via differentiable physics at training time yield significant improvements, as shown in Fig. 4f and 4g. While training with a few (1 to 4) steps yields improvements of up to 40% for the buoyancy-driven flow scenario, this number can be raised significantly by increasing the look-ahead at training time. A performance of more than 54% can be achieved by 64 recurrent solver iterations, while raising the look-ahead to

Table 1: A summary of the quantitative evaluation for the five PDE scenarios. $\text{SOL}_s$ denotes a variant with shorter look-ahead compared to SOL. (* For the CG solver scenario, iterations to reach an accuracy of 0.001 are given. Here, $\text{SOL}_s$ denotes the physics-based loss version.)

| Exp. | Mean absolute error of velocity | | | | | Rel. improvement | | | |
|------|------|------|------|------|------|------|------|------|------|
| | SRC | PRE | NON | $\text{SOL}_s$ | SOL | PRE | NON | $\text{SOL}_s$ | SOL |
| Wake Flow | 0.146±0.004 | 0.031±0.010 | 0.049±0.012 | 0.041±0.009 | 0.013±0.003 | 79% | 67% | 72% | 91% |
| Buoyancy | 1.590±1.033 | 1.373±0.985 | 1.080±0.658 | 0.944±0.614 | 0.620±0.390 | 19% | 29% | 41% | 60% |
| Adv.-diff. | 0.248±0.019 | 0.218±0.017 | 0.159±0.015 | 0.152±0.015 | 0.158±0.017 | 12% | 36% | 39% | 36% |
| *CG Solver | 121.6±13.44 | - | - | 79.03±10.02 | 29.59±14.83 | - | - | 35% | 76% |
| 3D Wake | 0.167±0.061 | - | 0.144±0.074 | - | 0.130±0.058 | - | 14% | - | 22% |

128 yields average improvements of 60%. Our tests consistently show that, without changing the number of weights or the architecture of a network, the gradients provided by the longer rollout times allow the network to anticipate the behavior of the physical system better and react to it. Throughout our tests, similar performances could not be obtained by other means.

**Generalization**  The buoyancy scenario also highlights the very good generalizing capabilities of the resulting models. All test simulations were generated with an out-of-distribution parametrization of the initial conditions, leading to substantially different structures, and velocity ranges over time.

**Training with Noise**  An interesting variant to stabilize physical predictions in the context of Graph Network-based Simulators was proposed by Sanchez et al. [48]. They report that perturbations of input features with noise lead to more stable long-term rollouts. We mimic this setup in our Eulerian setting by perturbing the inputs to the neural networks with $\mathcal{N}(0, \sigma)$ for varying strengths $\sigma$. While a sweet spot with improvements of 34.5% seems to exist around $\sigma = 10^{-4}$, the increase in performance is small compared to a model with less perturbations (30.6%), as training with an increased look-ahead for the SOL models gives improvements up to 60.0%.

**Training Stability**  The physical models we employ introduce a large amount of complexity into the training loop. Especially during the early stages of training, an inferred correction can overly distort the physical state. Performing time integration via the PDE then typically leads to exponential increases of existing oscillations and a diverging calculation. Hence, we found it important to pre-train networks with small look-aheads (we usually use $\text{SOL}_2$ models), and then continue training with longer recurrent iterations for the look-ahead. While this scheme can be applied hierarchically, we saw no specific gains from, e.g., starting a $\text{SOL}_{32}$ training with a $\text{SOL}_2$ model versus a $\text{SOL}_{16}$ model.

**Runtime Performance**  The training via differentiable physics incurs an increased computational cost at training time, as the PDE model has to be evaluated for $n$ steps for each learning iteration, and the calculation of the gradients is typically of similar complexity as the evaluation of the PDE itself. However, this incurs only moderate costs in our tests. For example, for the buoyancy-driven flow, the training time increases from 0.21 seconds per iteration on average for $\text{SOL}_2$ to 0.42s for $\text{SOL}_4$, and 1.25s for $\text{SOL}_{16}$. The look-ahead additionally provides $n$ times more gradients at training time, and the inference time of the resulting models is not affected. Hence, the training cost can quickly pay off in practical scenarios by yielding more accurate results without any increase in cost at inference time.

Computing solutions with the resulting hybrid method which alternates PDE evaluations and ANN inference also provides benefits in terms of evaluation performance: A pre-trained, fully convolutional CNN has an $\mathcal{O}(n)$ cost for $n$ degrees of freedom, in contrast to many PDE-solvers with a super-linear complexity. For example, a simulation as shown in Fig. 1 involving the trained model took 13.3s on average for 100 time steps, whereas a CPU-based reference simulation required 913.2s. A speed-up of more than $68\times$.

## 6  Conclusions

We have demonstrated how to achieve significant reductions of numerical errors in PDE-solvers by training ANNs with long look-ahead rollouts and differentiable physics solvers. The resulting models yield substantially lower errors than models trained with pre-computed data. We have additionally provided a first thorough evaluation of different methodologies for letting PDE-solvers interact with recurrent ANN evaluations.

Identical networks yield significantly better results purely by having the solver in the learning loop. This indicates that the numerical errors have regular structures that can be learned and corrected via learned representations. The resulting networks likewise improve generalization for out-of-distribution samples and provide stable, long-term recurrent predictions. Our results have the potential to enhance learning physical priors for a variety of deep learning tasks. Beyond engineering applications and medical simulations, a particularly interesting application of our approach is weather prediction [45], where a simple differentiable solver could be augmented with a learned correction function to recover the costly predictions of operational forecasting systems.

Overall, we hope that the demonstrated gains in accuracy will help to establish trained neural networks as components in the numerical toolbox of computational science.

## Broader Impact

PDE-based models are very commonly used and can be applied to a wide range of applications, including weather and climate, epidemics, civil engineering, manufacturing processes, and medical applications. Our work has the potential to improve how these PDEs are solved. As PDE-solvers have a long history, there is a wide range of established tools, some of which still use COBOL and FORTRAN. Hence, it will not be easy to integrate deep learning methods into the existing solving pipelines, but in the long run, our method could yield solvers that compute more accurate solutions with a given amount of computational resources.

Due to the wide range of applications of PDEs, our methods could also be used in the development of military equipment (machines and weapons) or other harmful systems. However, our method shares this danger with all numerical methods. For the discipline of computational science as a whole, we see more positive aspects when computer simulations become more powerful. Nonetheless, we will encourage users of our method likewise to consider ethical implications when employing PDE-solvers with learning via differentiable physics.

## Acknowledgments and Disclosure of Funding

This work is supported by the ERC Starting Grant *realFlow* (StG-2015-637014).

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
