[Supplementary Material]

# Appendix for *Solver-in-the-Loop: Learning from Differentiable Physics to Interact with Iterative PDE-Solvers*

**Kiwon Um**[1,2]    **Robert Brand**[1]    **Yun (Raymond) Fei**[3]    **Philipp Holl**[1]    **Nils Thuerey**[1]

[1]Technical University of Munich, [2]LTCI, Telecom Paris, IP Paris, [3]Columbia University

kiwon.um@telecom-paris.fr, robert.brand@tum.de
yf2320@columbia.edu, {philipp.holl, nils.thuerey}@tum.de

Below, we give additional details regarding the steps and numerical methods employed in each of the interaction variants discussed in the main text. We present details of the simulation setups for the five scenarios and give more detailed results for each case. Lastly, we discuss performance and list details of our neural network architectures.

As our experiments in the main text already demonstrate, deep learning algorithms that can closely interact with a differentiable PDE solver can yield substantially improved performance. This illustrates how crucial it is for deep learning algorithms that co-exist or interact with numerical solvers in a recurrent manner to anticipate shifts in the distributions of input features. We present additional results and show how interactions between PDE solvers and deep neural networks can be formulated. These interactions help to bridge the gap between distribution shifts that exist between different discretizations of a PDE. We will demonstrate that avoiding distribution shifts is essential for a model to infer a correction successfully. In our iterative setting, this, in turn, helps to keep the distributions aligned over the course of many iterations.

## A    Correction Functions for PDEs

For completeness, we provide a brief summary of our notation. We consider reference solutions $\mathbf{r}$ of the PDE $\mathcal{P}$ that are contained in the phase space manifold $\mathscr{R}$ with reference trajectories over time denoted by $\{\mathbf{r}_t, \mathbf{r}_{t+1}, \cdots, \mathbf{r}_{t+k}\}$ for $k$ steps of size $\Delta t$. A more coarsely approximated solution of the same problem is denoted by $s$ in the manifold $\mathscr{S}$ with trajectories $\{\boldsymbol{s}_t, \boldsymbol{s}_{t+1}, \cdots, \boldsymbol{s}_{t+k}\}$. We typically initialize the source state from the reference version via a transfer operator $\mathcal{T}$ with $\boldsymbol{s}_t = \mathcal{T}\mathbf{r}_t$ as initial condition. A transfer from source to reference states is denoted by $\mathcal{T}^T$.

The learning objective is to find the best possible correction function $\mathcal{C}(\boldsymbol{s} \,|\, \theta)$ given the weights $\theta$ and a network architecture. Without loss of generality, we assume that the correction function is applied additively, i.e., $\tilde{\boldsymbol{s}} = \boldsymbol{s} + \mathcal{C}(\boldsymbol{s} \,|\, \theta)$, where the tilde in $\tilde{\boldsymbol{s}}$ indicates the corrected state. A new state is computed in combination with the PDE via $\tilde{\boldsymbol{s}}_{t+1} = \mathcal{P}_s(\boldsymbol{s}_t) + \mathcal{C}(\mathcal{P}_s(\boldsymbol{s}_t) \,|\, \theta)$ for which we use the short form $(\mathcal{P}_s\mathcal{C})(\boldsymbol{s}_t)$ below. Multiple recurrent evaluations of a function are denoted by $\tilde{\boldsymbol{s}}_{t+k} = (\mathcal{P}_s\mathcal{C})^k(\boldsymbol{s}_t)$ for $k$ steps starting from an unaltered source state $\boldsymbol{s}_t$.

For training neural networks, we use an $L^2$-based loss, i.e., $\mathcal{L}(\tilde{\boldsymbol{s}}_t, \mathcal{T}\mathbf{r}_t) = \|\tilde{\boldsymbol{s}}_t - \mathcal{T}\mathbf{r}_t\|^2$, which is typically evaluated for $n$ steps via $\sum_{i=t}^{t+n} \mathcal{L}(\tilde{\boldsymbol{s}}_i, \mathbf{r}_i)$ in order to find a solution to the minimization problem: $\arg\min_\theta \sum_{i=t}^{t+n} \mathcal{L}(\tilde{\boldsymbol{s}}_i, \mathbf{r}_i)$.

We consider constrained advection-diffusion PDEs: $\partial\boldsymbol{u}/\partial t = -\boldsymbol{u} \cdot \nabla\boldsymbol{u} + \nu\nabla \cdot \nabla\boldsymbol{u} + \mathbf{g}$ subject to $\boldsymbol{M}\boldsymbol{u} = 0$. Here, $\boldsymbol{u}$, $\nu$, and $\mathbf{g}$ denote velocity, diffusivity, and external forces, respectively. The constraint matrix $\boldsymbol{M}$ contains an additional set of equality constraints imposed on $\boldsymbol{u}$.

## A.1 Learning Without Interaction

In the main text, we use learning via non-interacting trajectories as a baseline learning setup. In this case, a model is trained to minimize differences between states $s$ and $\mathbf{r}$ in a fully supervised manner. These versions are denoted by NON.

Despite its simplicity, different variants of this learning setup can be considered. In the simplest case, we initialize the source simulation from the corresponding reference version, evaluate the PDE once, and then train a model via a large number of such cases. In our notation, this means learning from states computed as $s_{t+1} = \mathcal{P}_s(\mathcal{T}\mathbf{r}_t)$. This effectively takes into account only a single evaluation of the source PDE, and a model can only learn from numerical differences that build up within this single step. Hence, a variant of this approach is to allow reference and target version to evolve over the course of multiple steps such that the errors in the source states $s$ show up more clearly with respect to $\mathbf{r}$. Similar to the look-ahead discussed in the main text, we can use $s_{t+n} = \mathcal{P}_s^n(\mathcal{T}\mathbf{r}_t)$ as a training data set. We denote such versions that have no interaction but consider multiple steps of unaltered coarse evolution as $\text{NON}_n$ below. Note that the previously discussed NON version could be denoted by $\text{NON}_1$, but we keep the label NON for consistency with the main text in the following.

For all choices of $n$, we obtain the following minimization problem for learning via non-interacting solvers:

$$\arg\min_{\theta} \sum_{i=0}^{n} \| s_{t+i} + \mathcal{C}(s_{t+i} \,|\, \theta) - \mathcal{T}\mathbf{r}_{t+i}\|^2. \tag{1}$$

Another non-interacting variant could be trained by reversing the setup above and initializing reference trajectories from source states, i.e., $\mathbf{r}_{t+n} = \mathcal{P}_R^n(\mathcal{T}^T s_t)$. Like before, a model could be trained in a supervised fashion from a data set of $s$ and $\mathbf{r}$ states computed in this way. However, as the interesting structures that make up the reference solutions typically take very long time spans to form (if they are achievable at all), this variant is clearly sub-optimal. Hence, due to the poor performance of the $\text{NON}_n$ versions, we have not included this reversed NON variant in our experiments.

The NON models presented in the main text so far already allow for a first quantification of the problems caused by the distribution shifts of the input features: across the two-dimensional fluid flow cases, the unaltered source simulations deviate by more than 50% in terms of MAE from the corrected simulations. This means that, after applying the corrections, the model receives inputs that strongly differ from those seen at training time. In terms of content of the input feature vectors, the MAE measurements show a change of over 50%. Nonetheless, we expect the model to reconstruct the reference states despite receiving inputs that are significantly different from the inputs seen at the time of training. Not surprisingly, the models only have limited success achieving this goal.

## A.2 Pre-computed Interactions

As an improvement over the non-interacting versions above, we consider a class of models learning from data generated via pre-computed interactions, denoted by PRE. The pre-computations have the goal of reducing the gap between source and reference trajectories. The pre-computation changes the source trajectories and thus provides the learning optimization with modified inputs that are closer to the reference at inference time. This scenario is common practice, e.g., for weather predictions, where simulations need to be aligned with real-world measurements, i.e., *data assimilation* algorithms [9, 17, 19]. As the data set has to be prepared only once, computationally expensive pre-computation is often still feasible as this overhead will not influence the performance at inference time. However, in the context of machine learning, pre-computed corrections can only provide limited improvements as the correction during the pre-computation phase can only partially mimic the behavior of the actual, learned version.

For PRE models, two correction functions are used: one for preparing the training data set denoted by $\mathcal{C}_{\text{pre}}$ and the learned correction $\mathcal{C}$. The training data set is computed as $\tilde{s}_{t+n} = (\mathcal{P}_s \mathcal{C}_{\text{pre}})^n (\mathcal{T}\mathbf{r}_t)$, where $n$ denotes the number of steps for independent simulation trajectories in the source and reference manifolds. Note that, in this context, due to the corrections being applied at the time of data generation, there is hope for longer unrolling periods (i.e., larger $n$) to have a positive effect on the learning outcome (in contrast to the $\text{NON}_n$ versions above). At inference time, $\mathcal{C}_{\text{pre}}$ is no longer used, and trajectories are instead computed as $\tilde{s}_{t+n} = (\mathcal{P}_s \mathcal{C})^n (s_t)$, in line with the NON variants. Hence, in total, four versions of a trajectory from a single initial phase space point $\mathbf{r}_t$ exist: a source

trajectory, a source trajectory corrected by pre-computation via $\mathcal{C}_{\text{pre}}$, a source trajectory corrected by the learned correction function $\mathcal{C}$, and the reference trajectory.

We first describe how to include a pre-computation correction for spatial corrections while taking into account simulation constraints before including the temporal dimension. For both, we adopt a constrained version of *best linear unbiased estimates* [6], which are widely used for data assimilation.

### A.2.1    Pre-computed Spatial Regularization

For a constraint-aware interpolation that can serve as a correction operator, consider two vector spaces $\mathbf{R} \in \mathbb{R}^{\chi}$ and $\mathbf{S} \in \mathbb{R}^{\xi}$ with different dimensionalities $\xi, \chi \in \mathbb{N}$ with $\xi < \chi$. Both vector spaces satisfy the constraint $M$, i.e., $M\mathbf{r} = 0$ for $\forall \mathbf{r} \in \mathbf{R}$, and $M\mathbf{s} = 0$ for $\forall \mathbf{s} \in \mathbf{S}$. Given a finer vector field $\mathbf{c}_R$, e.g., containing the reference solutions, we aim to find the closest vector field $\mathbf{c}_S$ ($\in \mathbf{S}$) to $\mathbf{c}_R$ ($\in \mathbf{R}$). Consider an interpolation operator $W$ that introduces new data points within a vector field $\mathbf{c}_S$ ($\in \mathbf{S}$), i.e., $W\mathbf{c}_S \in \mathbb{R}^{\chi}$. We, then, strive to minimize the distance between $W\mathbf{c}_S$ and $\mathbf{c}_R$ such that $\mathbf{c}_S$ can best represent the information of $\mathbf{c}_R$ without violating the constraints. Thus, we aim for computing $\mathbf{c}_S$ with

$$\arg\min_{\mathbf{c}_S} ||W\mathbf{c}_S - \mathbf{c}_R||^2 \quad \text{subject to} \quad M\mathbf{c}_S = 0. \tag{2}$$

This represents a constrained optimization problem with equality constraints, which we can solve via Lagrange multipliers $\lambda$ as follows:

$$\Phi = ||W\mathbf{c}_S - \mathbf{c}_R||^2 + (M\mathbf{c}_S)^\top \lambda. \tag{3}$$

This results in a system of equations:

$$\begin{bmatrix} W^\top W & -M \\ -M^\top & 0 \end{bmatrix} \begin{bmatrix} \mathbf{c}_S \\ \lambda \end{bmatrix} = \begin{bmatrix} W^\top \mathbf{c}_R \\ 0 \end{bmatrix}. \tag{4}$$

Using the Schur complement, we can simplify this system to speed up calculations:

$$M^\top (W^\top W)^{-1} M \lambda = M^\top (W^\top W)^{-1} W^\top \mathbf{c}_R, \tag{5}$$

$$\mathbf{c}_S = (W^\top W)^{-1} (W^\top \mathbf{c}_R - M \lambda). \tag{6}$$

In our setting, given source states $s$ and reference states $\mathbf{r}$, we can thus compute a correction vector field via $\mathbf{c}_t = (W^\top W)^{-1} (W^\top (\mathbf{r}_t - W s_t) - M \lambda)$, e.g., using $M = (\nabla \cdot)$ for Navier-Stokes scenarios. In order to train a model $\mathcal{C}(s \,|\, \theta)$ to infer the corrections, we can directly use the precomputed correction vectors:

$$\arg\min_{\theta} \sum_{i=0}^{n} ||\mathbf{c}_{t+i} - \mathcal{C}(\tilde{s}_{t+i} | \theta)||^2. \tag{7}$$

We will denote versions using this pre-computation scheme for $\mathcal{C}_{\text{pre}}$ with spatial regularization as $\text{PRE}_{\text{SR}}$.

### A.2.2    Pre-computed Spatiotemporal Regularization

The vector fields we target are obtained from a numerical simulation, where the underlying PDE is solved for a finite number of steps from an initial condition. In the context of deep learning, an important aspect to consider is the sensitivity [10] of the targeted function (i.e., the correction) with respect to the data at hand, i.e., in our case, the state of a source simulation. The pre-computation process described in the previous section is typically done on a per-time-step basis, and hence correction vector fields can vary significantly even for smooth changes of the source simulation. That means the correction function can have a very nonlinear and difficult to learn relationship with the observable data in a simulation.

In order to address this difficulty, we include a temporal regularization by limiting the changes over time for each sample point in space. Consequently, we regularize our correction vector fields such that they change smoothly in time by penalizing temporal change of the correction vector field within the Lagrange multiplier framework. We minimize $d\mathbf{c}_S/dt$ together with the constrained transfer from fine to coarse discretizations:

$$\arg\min_{\mathbf{c}_S} \left( ||W\mathbf{c}_S - \mathbf{c}_R||^2 + \beta ||\frac{d\mathbf{c}_S}{dt}||^2 \right) \quad \text{subject to} \quad M\mathbf{c}_S = 0. \tag{8}$$

Here, $\beta$ is the temporal regularization coefficient. A finite difference approximation of the temporal derivative of the correction field, i.e., $d\mathbf{c}_S/dt$, yields the following system of equations:

$$\begin{bmatrix} \boldsymbol{W}^\top \boldsymbol{W} + \beta \frac{2}{\Delta t} \mathbf{I} & -\boldsymbol{M} \\ -\boldsymbol{M}^\top & 0 \end{bmatrix} \begin{bmatrix} \mathbf{c}_S \\ \lambda \end{bmatrix} = \begin{bmatrix} \boldsymbol{W}^\top \mathbf{c}_R + \beta \frac{2}{\Delta t} \mathbf{c}_S^{t-1} \\ 0 \end{bmatrix}, \tag{9}$$

where $\Delta t$ is the time step size, $\mathbf{I}$ is the identity matrix, and $\mathbf{c}_S^{t-1}$ denotes the correction vector field evaluated at the previous time step. Following Eq. 7, this data is pre-computed and used for training a neural network in a supervised manner. Models trained with data from this spatiotemporal pre-computation as $\mathcal{C}_{\text{pre}}$ are denoted by PRE, and we have used a coefficient of $\beta = 1.0$ for all PRE models of our submission.

## A.3 Solver-in-the-Loop Interactions via Differentiable Physics

The main goal of training via differentiable physics is to bridge the gap that arises from changes in the input data distribution and directly train with the environment that the learned model is supposed to work with at inference time. Hence, the learning process aims to solve the minimization problem

$$\arg\min_{\theta} \sum_{i=0}^{n-1} \|\mathcal{P}_s(\tilde{s}_{t+i}) + \mathcal{C}(\mathcal{P}_s(\tilde{s}_{t+i})|\theta) - \mathcal{T}\mathbf{r}_{t+i+1}\|^2, \tag{10}$$

where the phase space trajectories are computed via $\tilde{s}_{t+k} = (\mathcal{P}_s\mathcal{C})^k(\mathcal{T}\mathbf{r}_t)$. This formulation illustrates that a cyclic dependency between the corrected states $\tilde{s}$ and the learned correction function $\mathcal{C}$ exists for the "solver-in-the-loop" interactions of this section. As both the deep neural network for $\mathcal{C}$ and likewise the PDE $\mathcal{P}_s$ are potentially highly non-linear operators, the corresponding coupled minimization problem for calculating the weights of $\mathcal{C}$ is challenging. However, our results clearly show that stable optimizations can be achieved in practice and that they lead to very significant improvements of the learned representation.

The recurrent training requires differentiable physics solvers that allow for a back-propagation of gradients through the discretized physical simulation. In this work, we employ a differentiable PDE solver from the open source $\Phi_{\text{Flow}}$ library [8]. This solver builds on the automatic differentiation of the underlying machine learning framework to compute analytic derivatives and augments them with custom derivatives where necessary. For example, the pressure correction step of a Navier-Stokes solver is provided with a custom gradient for performance reasons. This setup allows for a straightforward integration of solver functionality into machine learning models and enables end-to-end training in recurrent settings. Although all of our examples use the $\Phi_{\text{Flow}}$ solver, we do not leverage any special functionality apart from gradients being provided for all steps of the PDE solve. Hence, our results should carry over to other types of differentiable physics solvers.

It is worth noting that, in the setup discussed so far, the reference solver does not need to be differentiable; i.e., the phase space points in $\mathcal{R}$ could be provided by a black-box approximation without gradients as long as a differentiable solver for the source manifold $\mathcal{S}$ exists. We demonstrate the split setup using an external solver for the buoyancy-driven flows below.

Our implementation directly follows Eq. 10. For each mini-batch, we start with a collection of reference states $\mathbf{r}$ for which recurrent trajectories of $(\mathcal{P}_s\mathcal{C})^n$ are unrolled for $n$ steps. The loss with respect to corresponding reference states is computed over all intermediate states of the trajectory. Back-propagation, then, unrolls the differences through the sequence of solver steps to update the weights of the neural network that provides the correction function.

Under the assumption that the training process converges, this entirely removes the problem of distribution shift. Once the learned correction $\mathcal{C}$ converges to a steady-state, it is trained with exactly the phase state inputs that are produced at inference time. The MAE of the test data samples again provides a measure of the discrepancies. Compared to the differences of around 50% for non-interacting variants (measured between source states and corrected states), the deviations grow to 75% and above for SOL versions. Nonetheless, even this larger difference in terms of input distributions is unproblematic here as the network receives the modified states at training time. However, we noticed that, during our training runs, the final states typically do not fully converge, but still show smaller oscillations in terms of performance. While this could be prevented via learning-rate decay, we believe the slightly changing states provide robustness similar to dropout or manual injections of noise [13].

While the error accumulates and typically grows over the course of a full trajectory, our key hypothesis here is that a learned approach can nonetheless identify and correct a large part of the error function based on information from a single phase-space input. For the PDEs we consider, a single state uniquely describes its future evolution. We have experimented with additionally providing varying numbers of previous states $\mathbf{s}_{t-k}, ..., \mathbf{s}_{t-1}$ as input to our model. Our tests have not shown improvements from these additional states and indicate that the components of the error function that are learned with our approach can be reliably inferred from a single state $\mathbf{s}_t$.

## B  Experiments

To acquire our data sets, we generate a set of simulation sequences with varying initial conditions. These sequences are used for obtaining pairs of source and reference velocity fields for training. The following PDEs typically work with a continuous velocity field $\boldsymbol{u}$ with $d$ dimensions and components, i.e., $\boldsymbol{u}(\boldsymbol{x}, t) : \mathbb{R}^d \rightarrow \mathbb{R}^d$. For discretized versions below, $d_{i,j}$ will denote the dimensionality of a field such as the velocity with $i \in \{s, r\}$ denoting source/inference manifold and reference manifold, respectively. This yields $\boldsymbol{s} \in \mathbb{R}^{d \times d_{s,x} \times d_{s,y} \times d_{s,z}}$ and $\mathbf{r} \in \mathbb{R}^{d \times d_{r,x} \times d_{r,y} \times d_{r,z}}$ with domain size $d_x, d_y, d_z$ for source and reference. Typically, $d_{r,i} > d_{s,i}$ and $d_z = 1$ for $d = 2$. For all PDEs, we use non-dimensional parametrizations as outlined below, and the components of the velocity vector are denoted by $x, y, z$ subscripts, i.e., $\boldsymbol{u} = (u_x, u_y, u_z)^T$ for $d = 3$.

The mapping function $\mathcal{T}$ denotes a projection to the source manifold by $\mathcal{T}\mathbf{r}_t$, and we assume that the transpose transforms to the reference manifold, i.e., $\mathcal{T}^T \boldsymbol{s}_t$. The mapping function is typically neither bijective nor unique, i.e., $\mathcal{T}^T \mathcal{T} \mathbf{r}_t \neq \mathbf{r}_t$, however, within this work, we are primarily concerned with retrieving projected references of the form $\mathcal{T}\mathbf{r}_t$. The potential null-space of $\mathcal{T}^T$ is an interesting topic for super-resolution approaches [3]. We found that a bi- or tri-linear spatial downsampling from reference to source space is efficient to compute and yields sufficient accuracy for the transfer in our experiments. In order to make comparisons with the source simulations easier, we visualize the projected reference solution, i.e., $\mathcal{T}\mathbf{r}_t$, in the following.

### B.1  Unsteady Wake Flow in Two Dimensions

For the unsteady wake flow setup, we use the incompressible Navier-Stokes equations for Newtonian fluids:

$$\frac{\partial u_x}{\partial t} + \boldsymbol{u} \cdot \nabla u_x = -\frac{1}{\rho}\nabla p + \nu \nabla \cdot \nabla u_x$$
$$\frac{\partial u_y}{\partial t} + \boldsymbol{u} \cdot \nabla u_y = -\frac{1}{\rho}\nabla p + \nu \nabla \cdot \nabla u_y \tag{11}$$
$$\text{subject to} \quad \nabla \cdot \boldsymbol{u} = 0,$$

where $\rho$, $p$, $\nu$, and $g$ denote density, pressure, viscosity, and external forces, respectively. The constraint, $\nabla \cdot \boldsymbol{u} = 0$, is particularly important and introduces additional complexity as it restricts motions to the space of divergence-free (i.e., volume preserving) motions. The flow is integrated over time with operator splitting, and pressure is solved implicitly with a Chorin projection [2]. The domain $\Omega$ has an extent of $1 \times 2$ with open boundary conditions and a velocity inflow $\boldsymbol{u}_{\text{in}} = (0, 1)^T$ at the bottom face of the domain. A circular obstacle with diameter of $0.1$ is located at position $(1/2, 1/2)^T$. For reference simulations, the domain is discretized with $d_{r,x} = 128$ and $d_{r,y} = 256$ cells using a staggered layout for the velocity components. The source domain instead contains $d_{s,x} = 32$ and $d_{r,y} = 64$ cells. Data sets from both contain sequences of 500 time steps each. For the training data, the viscosity coefficient $\nu$ is chosen to yield Reynolds numbers $\text{Re}_{\text{train}} \in \{97.7, 195.3, 390.6, 781.3, 1562.5, 3125.0\}$; i.e., there is a factor of more than 30 between smallest and largest Reynolds numbers in the training data. The test data set instead contains the Reynolds numbers $\text{Re}_{\text{test}} \in \{146.5, 293.0, 585.9, 1171.9, 2343.8\}$, which are denoted as $\times 1$, $\times 2$, $\times 4$, $\times 8$, and $\times 16$ below, respectively.

**Training Procedure**  The neural network of $\mathcal{C}$ is fully convolutional. It consists of five ResBlocks [5] with $5 \times 5$ kernels. The convolutional layers have two times 32 features per block (details of the architecture are given in App. D). Overall, the model has around 260k trainable parameters. In addition to the velocity, the model receives a constant field containing the Reynolds number in order to distinguish the different physical regimes.

With the Reynolds number range above, we generate 500 time steps as training data, which contain temporal dynamics with ca. eight vortex shedding cycles for each case, i.e., they cover a similar number of eddy turnover times. This leads to roughly 98 million cells of data in the reference trajectories, which are down-sampled to 6.1 million cells with lower resolution of the source data. Example flow fields are shown in Fig. 6.

All SOL models are trained with the differentiable physics solver for 99.8k iterations with a batch size of 3 and a learning rate of $10^{-4}$. The NON model uses the same training modalities replacing the differentiable PDE solver with the supervised loss of Eq. 1. On the other hand, all PRE models are trained in a supervised manner for 36k iterations with a batch size of 32 and initial learning rate of $10^{-3}$ that is lowered to $5 \times 10^{-7}$ over the course of the training. Here, we augment the training data via randomized horizontal flipping and use 5% of the training data as validation samples. To show the stability of training, we train three models for each case below with different random seeds.

**Results**  We present results for the unsteady wake flow scenario using models trained via different interaction methodologies and evaluate each model on the test set of Reynolds numbers $Re_{test}$. Each simulation is computed for 500 time steps using the source solver in combination with a correction from a trained neural network. Mean errors are computed in comparison to reference phase space states, i.e., $\mathcal{T}\mathbf{r}$. We compute the errors over the three trained models for each variant.

In this scenario, the NON model already leads to a significant reduction of the overall velocity error. While the regular source simulation (SRC) shows a MAE of 0.146 with respect to the projected reference states $\mathcal{T}\mathbf{r}$, the NON model reduces this error to 0.049. These errors (and the following measurements) are mean values for all five test Reynolds numbers, which were not seen at training time. The results are visualized in Fig. 1, and corresponding numeric values are given in Table 1.

The pre-computed variants improve on this behavior, roughly halving the remaining error. The pre-computed variant without temporal regularization (PRE$_{SR}$) gives a worse performance than the one with spatiotemporal regularization (PRE) but, nonetheless, fares better than the NON version.

Fig. 1 additionally shows results for different SOL versions trained with the solver-in-the-loop interaction. While the SOL$_4$ version fares better than NON, it is only roughly on par with PRE$_{SR}$. Increasing the number of look-ahead steps, however, increases the performance substantially with the SOL$_{32}$ model exhibiting a final MAE of only 0.013. Several visual examples of simulated flows from the five test cases used in these evaluations are shown in Fig. 7. It is visible that the SOL version matches the behavior of the reference solution much more closely.

We additionally break down the errors with respect to the different Reynolds numbers of the five cases in Fig. 2. Despite a factor of 16 between the Reynolds numbers, there is no significant decrease in performance across the different cases. Only the NON version exhibits slightly larger errors for higher Reynolds numbers. On the other hand, the performance is largely uniform for the SOL versions.

Due to the distinct vortex shedding characteristics of the flow, it is interesting to evaluate the flow field in terms of its frequency spectrum. As an example, Fig. 3 shows the $u_x$ velocity component over the course of 500 simulation steps at the center of domain, i.e., behind the obstacle, for one of our test data sets. We show the corresponding evaluation in Fig. 4. Interestingly, especially the PRE versions fare better in terms of frequency errors. Here the relatively expensive pre-computation step shows a performance gain. Nonetheless, the models trained via differentiable physics likewise learn to control the frequency behavior when training with a sufficient number of look-ahead steps as the SOL$_{32}$ model yields a substantially lower frequency error than the PRE model.

We additionally show results for a smaller model for a simpler sequential convolutional network with 57k trainable parameters in Fig. 5. The overall relative ordering of the interaction methods remains the same. The non-interacting method performs worse than pre-computation, which in turn is outperformed by the differentiable physics interaction. However, the overall performance is reduced, e.g., the NON model only reduces the error by ca. 30%. The SOL$_{16}$ version still outperforms the other versions. Overall, not surprisingly, the reduced weight count significantly reduces the representational capabilities of the neural networks and leads to a deteriorated performance. Nonetheless, training via interactions with differentiable physics is beneficial for inference performance.

To conclude, approximate solutions of the unsteady wake flow case can be corrected substantially by learned models, and especially training with differentiable physics in the loop yields significantly reduced errors in long simulated sequences. The SOL$_{32}$ version with a larger model reduces the

MAE with respect to the reference solution to less than 9% (on average) of the error induced by the source simulation.

(a) Velocity error for different models

(b) Velocity improvement (relative to SRC) for different models

Figure 1: Different models applied to five test cases over 500 time steps for the unsteady wake flow scenario. The SOL$_{32}$ reduces the error introduced by SRC by a factor of 11.2 on average.

(a) Velocity error per Reynolds number

(b) Velocity improvement (relative to SRC) per Reynolds number

Figure 2: Separate evaluations for five different test cases of the unsteady wake flow scenario.

Figure 3: $u_x$-velocity at the center of domain for one test data set (Re = $\times 4$).

(a) Frequency error for different models     (b) Frequency error per Reynolds number

Figure 4: Frequency-domain evaluation for the unsteady wake flow scenario. Shown for the five test cases over 500 time steps.

(a) Velocity error for different smaller models    (b) Smaller models per Reynolds number

Figure 5: Different models with a smaller network size (57k trainable weights) applied to five test cases over 500 time steps for the unsteady wake flow scenario.

Figure 6: An example sequence of the unsteady wake flow from the training data set for time steps $t \in \{50, 60, \cdots, 200\}$.

Table 1: Quantitative evaluation of different models for the unsteady wake flow scenario.

| Model | **MAE Velocity**, Mean (std. dev.) | | | | | | | |
|---|---|---|---|---|---|---|---|---|
| | SRC | NON | $PRE_{SR}$ | PRE | $SOL_4$ | $SOL_8$ | $SOL_{16}$ | $SOL_{32}$ |
| Regular | 0.146 | 0.049 | 0.036 | 0.031 | 0.041 | 0.031 | 0.023 | 0.013 |
| | (0.004) | (0.012) | (0.009) | (0.010) | (0.009) | (0.012) | (0.004) | (0.003) |
| Smaller | 0.146 | 0.092 | 0.083 | 0.059 | - | 0.042 | 0.035 | - |
| | (0.004) | (0.028) | (0.025) | (0.015) | - | (0.011) | (0.010) | - |
| $L^2$ **Error of** $u_x$ **Velocity in Frequency Domain** | | | | | | | | |
| | SRC | NON | $PRE_{SR}$ | PRE | $SOL_4$ | $SOL_8$ | $SOL_{16}$ | $SOL_{32}$ |
| Regular | 0.557 | 0.202 | 0.106 | 0.087 | 0.194 | 0.128 | 0.101 | 0.051 |
| Smaller | 0.557 | 0.275 | 0.244 | 0.158 | - | 0.093 | 0.155 | - |

Figure 7: Time steps of test cases for the unsteady wake flow for $t \in \{50, 60, \cdots, 200\}$: (a) Re = $\times 1$ and (b) Re = $\times 16$.

## B.2 Buoyancy-driven Fluid Flow

This scenario encompasses a volume of hot smoke rising in a closed container. The motion of the smoke volume is driven by buoyancy forces computed via a marker field that is passively advected in the flow, and which marks a region of fluid with lower density. Assuming a small relative change of density between the marker and the bulk, we compute the resulting forces with a Boussinesq model. Hence, this scenario is likewise based on the Navier-Stokes equations, but due to the additional coupled system, it leads to significantly more chaotic and complex behavior than the unsteady wake flow. In order to target solutions with complex motions, we do not explicitly solve for viscosity effects, but rely on the numerical viscosity inherent in the discretization. This yields the following PDE:

$$\frac{\partial u_x}{\partial t} + \boldsymbol{u} \cdot \nabla u_x = -\frac{1}{\rho}\nabla p, \quad \frac{\partial u_y}{\partial t} + \boldsymbol{u} \cdot \nabla u_y = -\frac{1}{\rho}\nabla p + \eta d$$

$$\text{subject to} \quad \nabla \cdot \boldsymbol{u} = 0, \quad \frac{\partial d}{\partial t} + \boldsymbol{u} \cdot \nabla d = 0 \,, \tag{12}$$

where $\eta$ denotes the buoyancy factor for the Boussinesq model.

We also use this scenario to demonstrate that the reference data can be computed by a discretization or algorithm that differs from the one used to compute the source trajectories. More specifically, we use second-order pressure projection scheme for the reference trajectory solutions [20], which was shown to lead to an improved conservation of energy [4]. In addition, we use a less dissipative advection scheme for the source and reference solvers [14].

The domain has an extend of $1 \times 2$ units, where the marker density is injected in the lower quadrant. The reference simulations use a staggered discretization with $d_{r,x} = 128$ and $d_{r,y} = 256$, while the source simulations use a domain with $d_{s,x} = 32$ and $d_{r,y} = 64$. We randomize the initial size of the marker volumes with circular shapes with a radius $r \sim \mathcal{U}(0.1, 0.25)$, where $\mathcal{U}$ denotes a uniform distribution. The training data set consists of 48 different initial conditions simulated for 1000 steps each. Several examples are shown in Fig. 11. For the test scenes, we change the initial marker distribution $d$ to obtain five simulations containing two circles with $r \sim \mathcal{U}(0.05, 0.1)$ and another five simulations with $r \sim \mathcal{U}(0.2, 0.3)$. Thus, we obtain ten test scenes, half of which have a reduced marker quantity compared to the training data and five with an increased quantity. As the $d$ determines the forces induced by the Boussinesq model, this leads to simulations that are slower and faster, respectively, than those in the training set.

**Training Procedure** The neural network architecture for $\mathcal{C}$ follows the one described above, but instead uses four ResBlocks with 16 features each and contains ca. 36k trainable weights. As both velocity $\boldsymbol{u}$ and marker $d$ determine the dynamics of the flow, the network receives both fields as input, but still only infers a correction for the velocity; i.e., $d$ is modified only via advection through $\boldsymbol{u}$, not directly by $\mathcal{C}$. All SOL and NON models are trained for 294k iterations with a batch size of 4 and a learning rate of $10^{-4}$. We evaluate the models on validation set with 5 simulations and 300 time steps drawn from the same initial marker distribution as the training data, and keep the model with the lowest validation loss.

To speed up the pre-computations, we only compute $\mathcal{C}_{\text{pre}}$ for cells $i, j$ in the domain with $d_{i,j} > 10^{-4}$ (we validate this choice below). The PRE variants of $\mathcal{C}$ are then trained on the resulting, regularized data for 300k iterations with a batch size of 32 using horizontal flipping as data augmentation.

**Results** We evaluate different models which are applied to 300 time steps of ten test conditions. Errors with respect to the reference solutions are computed and averaged across the resulting 3k phase field states. Numeric error values for the following tests can be found in Table 2.

We evaluate the different baseline versions (NON and PRE) in comparison to the source simulation (which underlies all other variants) and compare them to SOL versions with increasing look-ahead. The resulting errors and relative improvements are shown in Fig. 8 and given numerically in Table 2. It is apparent that the SOL versions yield very significant improvements over the other learned variants. Besides the velocity errors, we also provide an evaluation of the passively advected marker density $d$. This quantity is crucial for the dynamics of the flow, but cannot be influenced directly by the neural networks. Hence, it provides an additional view on how well the inferred corrections manage to reduce the numerical errors of the source simulation. The corresponding evaluation highlights that

both velocity and density improvements increase consistently with SOL variants that were trained with larger look-aheads. We also evaluate the different models in terms of kinetic energy of the flows. As the kinetic energy is agnostic to the direction of the flow, the residual errors of the different variants do not show up as clearly as in the other evaluations. However, while the density and kinetic energy improvements are smaller than those for the velocity fields, the $SOL_{128}$ model nonetheless clearly outperforms the other variants.

Visualized evolutions of several test simulations are shown in Fig. 12. Here, the bi-modal nature of the test data with smaller (b) and larger (a,c) initial marker density configurations is shown. The different initial conditions lead to smaller and larger average velocities and, hence, highlight that the trained model generalizes very well.

**Ablations**   An evaluation of different neural network architectures for the buoyancy-driven flows with $SOL_2$ interaction illustrates how improvements stagnate beyond a certain network size and depth. For example, a model with more than 100k weights and almost three times the size of the regular model only yields an improvement of 3.6%. Another increase by a factor of four only gives 0.3% improvement. The corresponding graphs can be found in Fig. 9. Decreasing the network size, on the other hand, yields a performance that is 8.7% lower or even more for the smallest model. This motivates our choice to focus on the architecture with 36k trainable parameters, which was used for all other test with the buoyancy-driven flows.

As discussed in the main text, we also evaluated a method proposed by Sanchez et al. [13] to perturb inputs to network with noise in order to stabilize predictions. This approach shares our goal to reduce the shift of distributions for the input data such that the trained networks can produce more reliable estimates as they encounter new inputs at inference time. However, in contrast to the Lagrangian graph-based physics predictions, the added noise did not lead to large gains in our context. We test a variety of trained $SOL_2$ networks for which noise was injected into the input features, i.e., cell-wise samples of velocity and marker density, from a component-wise normal distribution $\mathcal{N}(0, \sigma)$ with standard deviation $\sigma$.

Details of the results are visualized in Fig. 10. As can be seen in the results, there is only a slight positive effect across a wide range of different noise strengths. The networks with $\sigma \sim 10^{-4}$ show the best results. However, the improvements of up to 34.6% via noise perturbations are surpassed by the $SOL_n$ models, where the best one yields an improvement of 59.8%. We think that the gains of our interacting model compared to injecting noise come from the systematic improvements of the SOL training, which potentially provides more reliable inputs at training time than stochastic perturbations. The fully convolutional nature of the networks additionally provides regularization at training time.

We have also evaluated how sub-optimal choices for solver interactions affect the inference performance. We train several NON models that are allowed to evolve for $n$ time steps without interaction, while computing a regular $L^2$ loss via Eq. 1. These versions are denoted with $NON_{dn}$ for $n$ steps of diverging evolution. In addition, we evaluate a model $PRE_{SR}$ using a pre-computed interaction without temporal regularization (i.e., only spatial) and one version ($PRE_F$) that uses the full spatiotemporal regularization without a density threshold; i.e., it requires several times more pre-computation by solving the Lagrange-multiplier minimization for the full spatial domains. Especially, the $NON_{dn}$ variants perform badly and exhibit large errors, with $NON_{d8}$ significantly distorting the flow behavior, instead of improving it. The corresponding evaluations are visualized in Fig. 13. It is likewise apparent that the additional PRE variants deteriorate the ability of the ANNs to correct the numerical errors of the source simulations.

To summarize, despite the complexity of the buoyancy-driven flows and the difficult reference trajectories produced by a higher-order PDE solver, the numerical errors of the source simulation can be reduced very successfully by training with the solver in the training loop.

(a) Velocity error  (b) Marker advection error  (c) Kinetic energy error

(d) Velocity improvement  (e) Impr. of marker advection  (f) Impr. of kin. energy

Figure 8: Velocity, marker advection, and kinetic energy errors for different models, especially for different SOL versions with increasing look-ahead. In the second row, we show improvements relative to the source version SRC.

(a) Trainable weights  (b) Velocity error  (c) Marker error  (d) Velocity improvement

Figure 9: $SOL_2$ training with different architectures that strongly vary the number of trainable parameters (a). While the smaller two models lead to a clear drop in accuracy, the larger two architectures yield small gains despite the increased weight count.

(a) Velocity improvement  (b) Improvement of marker advection

Figure 10: Varying levels of noise injected into the input features for $SOL_2$ at training time. While values around $10^{-4}$ lead to slight positive effects, the improvements are negligible compared to those achievable by the SOL variants.

Figure 11: An example sequence of the buoyancy scenario from the training data set for time steps $t \in \{0, 25, \cdots, 375\}$.

Figure 12: Several time steps $t \in \{50, 60, \cdots, 200\}$ of three buoyancy-driven fluid flow test cases (a)-(c).

(a) Velocity error  (b) Density error  (c) Velocity improvement

Figure 13: A comparison of models trained with a variety of sub-optimal interaction schemes for the buoyancy scenario. $\text{NON}_{dn}$ allows non-interacting models to evolve and diverge over $n$ steps, while $\text{PRE}_{SR}$ employs only spatial regularization in the pre-computation. $\text{PRE}_F$ resembles PRE, but was trained without a density threshold. Especially, the changes relative to SRC in (c) highlight that the $\text{NON}_{dn}$ variants have a negative effect.

Table 2: Quantitative evaluation of models for the buoyancy-driven flow scenario. $\text{M}_{XS,S,L,XL}$ denote different model sizes, while $\sigma_{1,2,3}$ denote models trained with noise of $\sigma = 10^{-3,-4,-5}$.

| **Quantity** | **MAE Velocity**, Mean (std. dev.) | | | | | | | |
|---|---|---|---|---|---|---|---|---|
| | SRC | NON | PRE | $\text{SOL}_2$ | $\text{SOL}_{16}$ | $\text{SOL}_{32}$ | $\text{SOL}_{64}$ | $\text{SOL}_{128}$ |
| Velocity | 1.590 | 1.079 | 1.373 | 1.027 | 0.859 | 0.775 | 0.695 | 0.620 |
| | (1.032) | (0.658) | (0.985) | (0.656) | (0.539) | (0.482) | (0.420) | (0.389) |
| Marker $d$ | 0.677 | 0.499 | 0.579 | 0.484 | 0.430 | 0.419 | 0.401 | 0.391 |
| | (0.473) | (0.336) | (0.409) | (0.325) | (0.281) | (0.277) | (0.262) | (0.253) |
| | $\text{M}_{XS}$ | $\text{M}_S$ | $\text{M}_L$ | $\text{M}_{XL}$ | $\sigma_1$ | $\sigma_2$ | $\sigma_3$ | $\text{NON}_{d4}$ |
| Velocity | 1.228 | 1.193 | 0.982 | 0.969 | 1.070 | 1.056 | 1.078 | 3.196 |
| | (0.746) | (0.826) | (0.646) | (0.626) | (0.683) | (0.700) | (0.706) | (1.404) |
| Marker $d$ | 0.521 | 0.494 | 0.461 | 0.466 | 0.503 | 0.496 | 0.503 | 0.656 |
| | (0.352) | (0.349) | (0.313) | (0.318) | (0.341) | (0.339) | (0.345) | (0.426) |

## B.3 Forced Advection-Diffusion

In the forced advection-diffusion scenario, we target a PDE environment with a constant, randomized forcing term. This forcing continuously injects energy into the dissipative system and takes the form of a spectrum of parametrized bands of sine waves. In this scenario, we target Burgers' equation. It represents a well-studied advection-diffusion PDE:

$$\frac{\partial u_x}{\partial t} + \boldsymbol{u} \cdot \nabla u_x = \nu \nabla \cdot \nabla u_x + g_x(t), \quad \frac{\partial u_y}{\partial t} + \boldsymbol{u} \cdot \nabla u_y = \nu \nabla \cdot \nabla u_y + g_y(t), \tag{13}$$

where $\nu$ and $\mathbf{g}$ denote diffusion constant and external forces, respectively. Our setup resembles a 2D variant of the tests employed by the work on learning data-driven discretizations [1]; correspondingly, we extend the forcing terms described there to 2D. We generate the forces from 20 overlapping sine functions each with a random direction, amplitude, and phase shift:

$$g_x(t) = \sum_{i=1}^{20} \cos(\alpha_i) a_i \sin(\omega_i t - kx + \phi_i), \quad g_y(t) = \sum_{i=1}^{20} \sin(\alpha_i) a_i \sin(\omega_i t - kx + \phi_i). \tag{14}$$

This PDE scenario does not involve any equality constraints, i.e., $\boldsymbol{M} = 0$.

Similar to the previous scenarios, we discretize the system on a staggered grid and compute the advection operator with a semi-Lagrangian scheme [16]. The domain has a square, normalized size of $1 \times 1$ with reference trajectories computed via a resolution of $d_{r,x} = d_{r,y} = 128$. The source domain correspondingly uses $d_{s,x} = d_{s,y} = 32$.

**Training Procedure and Results** As training data, ten simulations of 200 steps each are used. An example sequence of the data is shown in Fig. 14. The SOL and NON models are trained for 38.4k steps with a batch size of five with a learning rate of $10^{-4}$, while the PRE model is trained for 25k steps with a batch size of 32 using an initial learning rate of $10^{-3}$ that was lowered to $5 \times 10^{-7}$ over the course of the training. The PRE model additionally uses 5% of the training data set for validation. The test data set contains five cases with different initial conditions and force fields over the course of 200 time steps. All models use a neural network architecture with five ResBlocks with 32 features each.

As summarized in the main text, the learned correction functions can significantly decrease the numerical errors of the source simulation. Across the different test cases (partly shown in Fig. 15), the best models achieve a reduction by over 67%. The corresponding MAE measurements are given in Table 3, and Fig. 16 provides an overview of the performance per test case. While the PRE model shows a lower performance, most likely due to an overly strong temporal regularization, the NON model is close to the best SOL model in this case with an MAE of 0.159 compared to 0.148 for SOL$_2$. Interestingly, this behavior matches the results of Bar-Sinai et al. [1]. They experimented with up to eight recurrent steps of a 1D Burgers' simulation, but did not report significant advantages from training with the 1D solver in the loop.

In contrast, we found that more interactions show their advantage in a deterministic scenario, where we exclude the external forces from the Burgers' equation above, i.e., Eq. 13. As this versions exhibits less chaotic behavior, the SRC version generally shows smaller errors compared to the SRC version in the forced scenario. The SOL versions now yield further improvements when trained with more look-ahead: SOL$_4$ yields an improvement of 10% over SRC, SOL$_{16}$ yields 12%, while the SOL$_{32}$ version reduces the error by 17%. Table 3 shows the corresponding MAE measurements.

Our results highlights that deep learning via physical simulations works particularly well when the ANNs can actually learn to predict the behavior of the dynamics and, thus, compensate for the numerical errors that will occur. If, on the other hand, external and unpredictable influences such as the randomized forcing terms dominate the behavior, the model has a reduced chance to predict the right correction function.

Table 3: Quantitative evaluation of different models for the forced advection-diffusion scenario. MAE values without forcing are given with a ×100 factor.

| **MAE Velocity**, Mean (std. dev.) | | | | | | |
|---|---|---|---|---|---|---|
| With forcing | SRC | PRE | NON | $SOL_2$ | $SOL_4$ | $SOL_8$ |
| | 0.248 (0.019) | 0.218 (0.017) | 0.159 (0.015) | 0.148 (0.016) | 0.152 (0.015) | 0.158 (0.017) |
| Without forcing | SRC | NON | $SOL_4$ | $SOL_8$ | $SOL_{16}$ | $SOL_{32}$ |
| (×100) | 0.306 (0.020) | 0.272 (0.028) | 0.276 (0.037) | 0.277 (0.040) | 0.268 (0.030) | 0.253 (0.020) |

Figure 14: An example sequence from the training data set of the forced advection-diffusion test case.

Figure 15: Time steps of three test cases (a)-(c) from the forced advection-diffusion scenario.

(a) Velocity error per test case

(b) Velocity improvement (relative to SRC) per test case

Figure 16: Separate evaluations for five different test cases of the forced advection-diffusion scenario.

## B.4 Inference of Initial Guesses for Conjugate Gradient Solvers

In this section, we investigate the interaction of learning models with conjugate gradient (CG) solvers [7]. We target Poisson problems, which often arise many PDEs, e.g., in electrostatics or in fluid flow problems where the pressure $p$ is computed via $\nabla \cdot \nabla p = \nabla \cdot \boldsymbol{u}$. Specifically, we explore the iteration behavior of the CG solver given an initial state predicted by a trained model. To this end, we compare three main methods: A solver-in-the-loop ($\text{SOL}_n$) approach, a non-interacting supervised approach (NON), and a differentiable physics-based ($\text{SOL}_{\text{DIV}}$), which is trained to directly minimize the PDE residual. In general, the CG solver iterations converge toward a reference pressure field $p$ such that $\boldsymbol{A}p = \nabla \cdot \boldsymbol{u}$ with $\boldsymbol{A} = \nabla \cdot \nabla$. For an intermediate solution $\hat{p}$, the residual $r = \nabla \cdot \boldsymbol{u} - \boldsymbol{A}\hat{p}$ measures how far away the approximated pressure $\hat{p}$ is from the true solution. Thus, as the solver converges, $r$ decreases and the difference $\hat{p} - p$ converges to zero. In the following, we employ the neural network $\mathcal{C}$ to infer a pressure field given a velocity sample $\boldsymbol{u}$, i.e., $\hat{p} = \mathcal{C}(\boldsymbol{u})$. We focus on 2D cases, i.e., $\boldsymbol{u} \in \mathbb{R}^{2 \times d_x \times d_y}$ and $p, r \in \mathbb{R}^{d_x \times d_y}$.

**Loss Functions**   The NON version employs a regular supervised loss, i.e., the difference of the predicted pressure $\hat{p}$ from the pre-computed reference pressure $p$ for $j$ different samples:

$$\mathcal{L}_{\text{NON}} = \|\mathcal{C}(\boldsymbol{u}) - p\|^2. \tag{15}$$

We additionally compare to a variant that is often referred to as *unsupervised* in previous work, and which is in line with other physics-based or physics-informed loss constructions [11, 15]. Specifically, the $\text{SOL}_{\text{DIV}}$ version replicates the setup described in [18] and uses the PDE $\nabla^2 p - \nabla \cdot \boldsymbol{u} = 0$ as loss for the training of a neural network. Given an input velocity $\boldsymbol{u}^*$, the goal is to infer a pressure function $\hat{p}(\nabla \cdot \boldsymbol{u}^*)$ such that the PDE residual is minimized:

$$\mathcal{L}_{\text{SOL}_{\text{DIV}}} = \|\nabla \cdot \boldsymbol{u}^* - \nabla \cdot \nabla \mathcal{C}(\boldsymbol{u})\|^2. \tag{16}$$

This version represents a different form of differentiable PDE solvers, namely including them in the loss formulation, and hence we denote it with $\text{SOL}_{\text{DIV}}$. However, due to a lack of iterating calculations for this variant, a more appropriate name would be *"solver-in-the-loss"* rather than *"solver-in-the-loop"*.

As a third variant, we employ a solver-in-the-loop interaction that employs a differentiable CG solver and uses a learning objective to minimize the PDE residual after $n$ iterations of the CG solver. In this scenario, $\mathcal{P}_s$ represents a linear operator, i.e., one step of the CG method to approximate $\nabla^{-2}(\nabla \cdot \boldsymbol{u})$, and the loss function is given by:

$$\mathcal{L}_{\text{SOL}_n} = \|\mathcal{P}_s^n(\mathcal{C}(\boldsymbol{u})) - \mathcal{C}(\boldsymbol{u})\|^2. \tag{17}$$

Thus, the $\text{SOL}_n$ and $\text{SOL}_{\text{DIV}}$ both minimize the same residual divergence $r$; while the $\text{SOL}_{\text{DIV}}$ loss aims to do so directly, the $\text{SOL}_n$ version instead sees how the iterative solver performs. At training time, the $\text{SOL}_n$ variant receives gradients through $n$ iterations of the iterative solver via back-propagation.

**Training Procedure**   The trained models in this section all use the same convolutional U-net architecture [12] with 22 layers of strided convolutions and $5 \times 5$ kernels, containing around 127k trainable parameters (see App. D for details). The training data set was generated using the conjugate gradient solver from the $\Phi_{\text{Flow}}$ framework [8]. It is comprised of 3k fluid simulations on a domain with $d_x = d_y = 64$ and closed boundaries. Each simulation consists of a randomly generated density and velocity field, which are integrated over time for 16 steps. Each model was trained for 300k steps with a learning rate of $2 \times 10^{-4}$ and training batch size of 16. The reference solutions were pre-computed with a CG solver using an accuracy threshold of $10^{-6}$ for the residual norm.

**Results**   We now compare the different loss functions by their performance in conjunction with the CG solver. We compute averages for 100 test cases each time, i.e., samples that were not seen at training time. As baseline, we denote a CG solving process that starts from a zero guess as SRC.

We first compare how many CG iterations are required to reach a certain target accuracy given the inferred solutions by the three different types of models. The results are shown in Table 4 and visualized in Fig. 17. Initially, $\text{SOL}_{\text{DIV}}$ reaches an accuracy of almost $10^{-2}$, closely followed by $\text{SOL}_5$. While the supervised NON version produces pressure predictions that seem quite close to the

reference, its initial accuracy is only slightly better than the zero guess employed by SRC. This is due to the error being measured locally per grid point, while the correctness of larger structures becomes more important after in interactions with the CG solvers. Over the first five to ten CG iterations, the accuracy of $SOL_5$ improves very quickly, overtaking the other methods. To reach an accuracy of $10^{-2}$, the CG solver requires an average of around two steps in conjunction with $SOL_5$, nine steps with NON, 28 steps with $SOL_{DIV}$ and 78 steps starting from zero. When running the CG solver for more iterations, the accuracy increases similarly for all methods, with $SOL_5$ retaining its advantage.

Comparing $SOL_5$ to $SOL_{DIV}$ shows the importance of training with the solver in the loop: the $SOL_{DIV}$ model does not receive any feedback regarding the behavior of the solver. It predicts solutions that satisfy the loss – measured per grid point – but do not match the large-scale structures of the true solution. Consequently, this task is left to the CG solver, which requires many iterations to work out the correct global solution. The $SOL_5$ model, however, sees the corrections performed by the CG solver at training time and can learn to adjust its guess accordingly.

When investigating the inferred pressure fields themselves (Fig. 18), we see that the guesses of the $SOL_5$ model come closest to the reference, followed by those of the NON variant. The $SOL_{DIV}$ differs more strongly, and the residual divergence, shown in Fig. 19, highlights that it has a noticeable error pattern near the outer border of the domain. This provides an explanation for the poor behavior of the $SOL_{DIV}$ model for the initial CG solver iterations: while it minimizes the PDE-based loss in an absolute sense, it does not receive information about how different parts of the solution influence the future iterations of the solver. This ambiguity is alleviated to some extent by the pre-computed reference solutions for NON, but especially the $SOL_5$ version receives this feedback in terms of gradient from the differentiable solver and, in this way, can best adapt to the requirements for future iterations.

We also experimented with varying the number of look-ahead steps for $SOL_n$ models in the loss function of Eq. 17. This ablation study (Fig. 20) shows how too few iterations clearly deteriorate the performance, while more than 5 iterations lead to a slight increase in the required iterations. We assume that this behavior is potentially caused by evaluating the loss only for the final output of the $n$ iterations.

**Discussion** Our results highlight the advantages of training with the solver in the loop for fully implicit PDE solvers. Likewise, it shows that a physics-informed loss formulation alone yields only a partial view of the problem. While a loss-based residual cannot adapt to iterative algorithms, the solver-in-the-loop models directly receive gradient-based feedback at training time.

The combination of an inferred initial guess with a traditional solver represents a particularly interesting hybrid algorithm, as it gives convergence guarantees that a learned approach alone would not be able to provide. Even if a trained model generates a sub-optimal solution, the solver can improve the solution until it matches the desired accuracy threshold. On the other hand, pre-training a model for a known problem domain can significantly reduce the required number of iterations and, consequently, reduce the workload in scenarios where PDEs from the same problem domain need to be solved repeatedly and in large numbers. Here, the current hardware developments provide an additional promise: the advances in terms of highly specialized hardware for evaluating neural networks can provide a substantial future speed-up even for a fixed, pre-trained model.

Figure 17: (a) Iterations needed to reach target accuracy and (b) comparison of maximum residual error over iterations.

Figure 18: (a) Sample outputs of the models and (b) difference of output from reference.

Figure 19: Residual error after one CG solver iteration.

Figure 20: Comparison of SOL models with different look-ahead steps.

Table 4: Evaluation of the CG solver performance for different models.

| Model | **Iterations for Accuracy**, Mean (std. dev.) | | | | | |
|---|---|---|---|---|---|---|
| | $10^{-1}$ | $10^{-2}$ | $10^{-3}$ | $10^{-4}$ | $10^{-5}$ | $10^{-6}$ |
| NON | 1.67 | 9.33 | 52.16 | 109.12 | 155.37 | 186.12 |
| | (1.010) | (5.428) | (17.540) | (15.875) | (10.155) | (5.719) |
| $SOL_{DIV}$ | 0.0 | 27.79 | 79.06 | 117.97 | 155.76 | 181.07 |
| | (0.0) | (15.255) | (10.042) | (13.234) | (9.403) | (6.052) |
| $SOL_5$ | 0.03 | 1.97 | 29.59 | 88.37 | 133.59 | 167.37 |
| | (0.171) | (1.118) | (14.832) | (13.465) | (11.605) | (8.549) |

## B.5 Three-dimensional Unsteady Wake Flow

As a final scenario, we target a three-dimensional fluid flow problem. The third spatial dimension leads to a large increase in terms of degrees of freedom, especially in the finer reference manifold. Additionally, the three axes of rotation lead to significantly more complicated flow structures.

Overall, we target a setup that represents an extension of the 2D unsteady wake flow case of App. B.1. Instead of a circular obstacle, the flow now faces a cylindrical obstacle in a 3D domain with extent of $1 \times 1 \times 2$. The cylinder with diameter $0.1$ is located at position $(1/2, 1/2, 0)^T$ and has an extent of 1 unit along the z-axis. We use the incompressible Navier-Stokes equations in three dimensions as underlying PDE:

$$\frac{\partial u_x}{\partial t} + \boldsymbol{u} \cdot \nabla u_x = -\frac{1}{\rho}\nabla p + \nu \nabla \cdot \nabla u_x$$

$$\frac{\partial u_y}{\partial t} + \boldsymbol{u} \cdot \nabla u_y = -\frac{1}{\rho}\nabla p + \nu \nabla \cdot \nabla u_y$$

$$\frac{\partial u_z}{\partial t} + \boldsymbol{u} \cdot \nabla u_z = -\frac{1}{\rho}\nabla p + \nu \nabla \cdot \nabla u_z$$

$$\text{subject to} \quad \nabla \cdot \boldsymbol{u} = 0. \tag{18}$$

For reference simulations, the domain is discretized with $d_{r,x} = d_{r,y} = 128$ and $d_{r,z} = 256$ cells using a staggered layout for the velocity components. The source domain has a resolution of $d_{s,x} = d_{s,y} = 32$ and $d_{r,z} = 64$ cells. Data sets from both domains contain phase space trajectories of 500 time steps. For the training data, the viscosity coefficient $\nu$ is chosen to yield Reynolds numbers $\text{Re}_{\text{train}} \in \{58.6, 78.1, 117.2, 156.3, 234.4, 312.5, 468.8, 625.0\}$. While the range of Reynolds numbers covers a slightly reduced range compared to the 2D case, there is still a factor of more than ten between largest and smallest ones, and the 3D nature of the flow introduces a significant amount of complexity. The example visualizations of a training data set in Fig. 21 highlight the complexity of the flows.

For the test set, we use different Reynolds numbers, namely $\text{Re}_{\text{test}} \in \{68.4, 97.7, 195.3, 136.7, 273.4, 390.6, 546.9\}$. The following test evaluations were computed for the seven Reynolds numbers in $\text{Re}_{\text{test}}$ over 300 time steps. Numeric values are given in Table 5.

**Training Procedure** For the 3D case, we use a ResNet that largely follows the architecture of the 2D cases, but employs 3D convolutions instead. The ResNet contains six blocks with kernel sizes of $5 \times 5 \times 5$ and $3 \times 3 \times 3$ for the two convolutional layers per block. The number of filters is increases to 48 in the center of the network, yielding 1002k trainable parameters (also see App. D). As for the 2D case, the inputs for the 3D models contain a constant field indicating the targeted Reynolds number. All models were trained for 300k iterations using a learning rate of $10^{-4}$ and a batch-size of four. We then use three validation simulations with $\text{Re}_{\text{val}} \in \{61.0, 305.2, 470.0\}$ to select the best performing model.

Due to the increased computational workload to train the 3D models, we focus on a NON variant and a $\text{SOL}_{16}$ version, which uses the same differentiable Navier-Stokes solver for producing gradient information over the course of up to 16 unrolled simulation and inference steps for each iteration at training time. This version was trained with $\text{SOL}_8$ for 200k iterations and then for an additional 100k iterations as $\text{SOL}_{16}$.

**Results** The 3D flow represents a significant increase in terms of complexity for the deep learning models. Among others, we were not able to train a stable NON version despite numerous tests. While the models performed well for ca. 100 to 150 time steps, small scale oscillations induced by the corrections accumulate and start to strongly distort the flow. This is a good example of the undesirable shift of distributions for the inputs: once the phase space trajectories produced by the hybrid method leave the distribution of the regular source states seen at training time, the model fails to infer reasonable corrections.

In contrast, the $\text{SOL}_{16}$ version retains its stability over the course of long simulations with several hundred steps. This is reflected in the MAE measurements of the velocity fields over the test cases: the regular source simulation induces an error of 0.167, which the NON version reduces to 0.143. The $\text{SOL}_{16}$ reduces the error to 0.130 instead, which however only gives a partial view of the overall

behavior of the different versions. The graphs over time shown in Fig. 22a illustrate the diverging behavior of the NON version. While it does very well initially, even slightly surpassing $SOL_{16}$ around frame 100, the errors quickly grow afterwards, eventually leading to a performance that is worse than the source simulation.

The frequency graphs of the kinetic energy in Fig. 22b, measured for an array of $5^3$ sample points at the center of the domain, instead show that the $SOL_{16}$ simulations closely match the frequency distribution of the reference simulations. It succeeds in restoring the change of frequencies across the different temporal scales of the flow significantly better than the SRC and NON models. The source simulation instead underestimates larger frequencies and over-estimates smaller ones.

Fig. 23 visualizes the vorticity magnitude of several test cases with Reynolds numbers not seen during training. The $SOL_{16}$ model manages to correct the vortex shedding behavior of the source simulation and closely matches the reference. As we visualize in the supplemental video, the NON version starts to oscillate, injecting undesirable distortions into the velocity field.

Figure 21: Two example sequences with (a) Re=117.2 and (b) Re=273.4 of the three-dimensional wake flow from the training data set. Each row shows 200 time steps for SRC (top) and reference versions (bottom) in terms of vorticity magnitude.

(a) Velocity error over time

(b) Frequency error

Figure 22: Evolutions of velocity MAE and frequency errors over the course of 300 time steps averaged for the seven test cases of the three-dimensional wake flow. (a) The NON versions perform well initially, but strongly diverges for later frames. (b) The $SOL_{16}$ shows a clearly improvement in terms of the frequency distribution of the kinetic energies. The overall curve of $SOL_{16}$ closely follows the reference with an initial offset over the reference, which inherits from the source simulation.

Figure 23: Three test cases with (a) Re=68.4, (b) Re=136.7, and (c) Re=546.9. Each row shows time steps over the course of 200 time steps for SRC, $SOL_{16}$, and the reference (top to bottom). The $SOL_{16}$ model interacting with the source solver successfully preserves the complex rotating motions behind the cylindrical obstacle (middle), which the regular source solver cannot resolve (top).

Table 5: Quantitative evaluation of different models for the three-dimensional wake flow scenario.

| MAE Velocity, Mean (std. dev.) | | | Freq. MAE Kinetic Energy, Mean (std. dev.) | | |
|---|---|---|---|---|---|
| SRC | NON | $SOL_{16}$ | SRC | NON | $SOL_{16}$ |
| 0.167 (0.035) | 0.143 (0.070) | 0.130 (0.024) | 0.0614 (0.133) | 0.074 (0.209) | 0.058 (0.088) |

## C Performance

We measure the computational performance of our models in comparison to a reference simulation on a workstation with an Intel Xeon E5-1650 CPU with 12 virtual cores at 3.60GHz and an NVIDIA GeForce GTX 1080 Ti GPU. As reference solver, we employ a CPU-based simulator using OpenMP parallelization. We compare this with our (relatively un-optimized) differentiable physics framework, which evaluates the PDE and the trained model within *TensorFlow* on the GPU.

For the buoyancy-driven flow simulation, the CPU-based reference simulation requires 5.79 seconds on average for 100 time steps. Instead, evaluating the $SOL_{128}$ neural network model itself requires an accumulated 0.43 seconds. For comparison, computing 100 time steps of the source solver takes 0.476 seconds. In comparison to the inference for forward simulations with a pre-trained model, each iteration during training is significantly more expensive: for the $SOL_8$, $SOL_{16}$, and $SOL_{32}$ models of the 2D wake flow case, a training iteration took 0.6, 1.3, and 2.5 seconds on average, respectively. As

this is a one-time, pre-processing cost, the gains in performance of the resulting hybrid solver can quickly offset the computational expense for training a model.

The computational workload for PDE solvers typically rises super-linearly with the number of degrees of freedom. Hence, the gap is even more pronounced when considering the 3D wake flow case. Here, the reference simulation requires 913.2 seconds for 100 time steps, while the $SOL_{16}$ version requires 13.3 seconds on average. Thus, the source simulation with learned corrections is more than 68 times faster than the reference simulation.

Despite the substantial reduction in terms of runtime, we believe these performance results are preliminary, and far from the speed-up that could be achieved in optimal settings with a learning-augmented PDE solver. An inherent advantage of combining an approximate PDE solver with a deep-learning-based corrector ANN stems from the fact that a relatively simple solver suffices as a basis. Hence, while existing reference solvers in scientific computing fields might come with vast existing code-bases, the source solver could encompass only a small subset of the full solver and introduce the residual dynamics via a learned component. This would also reduce the work to provide gradients for the source solver, which many existing simulation frameworks do not readily offer. Due to its reduced scope, the source solver would also be significantly easier to optimize. Additionally, the learned corrector component would trivially benefit from all future hardware advances for efficient evaluations of neural networks. Hence, we believe that, in practice, a much more substantial speed-up will be achievable than the ones we have measured for the two- and three-dimensional simulation scenarios of this work.

## D  Neural Network Architectures

Below, we give additional details of the network architectures used for the five different scenarios. We intentionally slightly vary the architecture to demonstrate that our solver-in-the-loop approach does not rely on a single, specific architecture. We employ ResNets for the large majority of the PDE interaction models as the correction task resembles a translation from phase space input quantities to a field of localized corrections. The CG solver scenario, on the other hand, requires a more global view, which motivates our choice of a U-net architecture. The overall structure with kernel sizes and feature maps of both types of networks is illustrated in Fig. 24. We additionally list hyperparameters for each architecture in Table 6.

Figure 24: A visual summary of the two main architectures of the neural networks used for Sect. B.1 to B.3 (left), and Sect. B.4 (right).

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

Table 6: Hyperparameters of neural network architectures.

| Experiment | Arch. | Layers | Features | Conv. Kernels | Train. Weights |
|---|---|---|---|---|---|
| 2D Wake Flow B.1 | Res-Net | 12 | 32 | $5^2$ | 260,354 |
| 2D Wake Flow B.1, Small | Sequential | 3 | 32, 64 | $5^2$ | 56,898 |
| Buoyancy B.2, $M_{XS}$ | Res-Net | 6 | 4 | $5^2, 3^2$ | 1,310 |
| Buoyancy B.2, $M_S$ | Res-Net | 8 | 8 | $5^2, 3^2$ | 5,114 |
| Buoyancy B.2, Regular | Res-Net | 10 | 16 | $5^2, 3^2$ | 35,954 |
| Buoyancy B.2, $M_L$ | Res-Net | 14 | 16, 32 | $5^2, 3^2$ | 100,114 |
| Buoyancy B.2, $M_{XL}$ | Res-Net | 14 | 32, 64 | $5^2, 3^2$ | 400,930 |
| Forced Adv.-diff. B.3 | Res-Net | 12 | 32 | $5^2$ | 261,154 |
| CG Solver App. B.4 | U-Net | 22 | 4, 8, 16, 32 | $5^2$ | 127,265 |
| 3D Wake Flow B.5 | Res-Net | 14 | 24,48 | $5^3, 3^3$ | 1,002,411 |

[5] K. He, X. Zhang, S. Ren, and J. Sun. Deep residual learning for image recognition. *arXiv:1512.03385 [cs]*, Dec. 2015.

[6] C. R. Henderson. Best linear unbiased estimation and prediction under a selection model. *Biometrics*, pages 423–447, 1975.

[7] M. R. Hestenes, E. Stiefel, et al. Methods of conjugate gradients for solving linear systems. *Journal of research of the National Bureau of Standards*, 49(6):409–436, 1952.

[8] P. Holl, V. Koltun, and N. Thuerey. Learning to control pdes with differentiable physics. *International Conference on Learning Representations (ICLR)*, 2020.

[9] C. Jones and B. Macpherson. A latent heat nudging scheme for the assimilation of precipitation data into an operational mesoscale model. *Meteorological Applications*, 4(3):269–277, 1997.

[10] J. M. Murphy, D. M. Sexton, D. N. Barnett, G. S. Jones, M. J. Webb, M. Collins, and D. A. Stainforth. Quantification of modelling uncertainties in a large ensemble of climate change simulations. *Nature*, 430(7001):768, 2004.

[11] M. Raissi, A. Yazdani, and G. E. Karniadakis. Hidden fluid mechanics: A navier-stokes informed deep learning framework for assimilating flow visualization data. *arXiv:1808.04327*, 2018.

[12] O. Ronneberger, P. Fischer, and T. Brox. U-net: Convolutional networks for biomedical image segmentation. In *International Conference on Medical Image Computing and Computer-Assisted Intervention*, pages 234–241. Springer, 2015.

[13] A. Sanchez-Gonzalez, J. Godwin, T. Pfaff, R. Ying, J. Leskovec, and P. W. Battaglia. Learning to simulate complex physics with graph networks. *arXiv:2002.09405*, 2020.

[14] A. Selle, R. Fedkiw, B. Kim, Y. Liu, and J. Rossignac. An unconditionally stable maccormack method. *Journal of Scientific Computing*, 35(2-3):350–371, June 2008.

[15] J. Sirignano and K. Spiliopoulos. Dgm: A deep learning algorithm for solving partial differential equations. *Journal of Computational Physics*, 375:1339–1364, 2018.

[16] J. Stam. Stable fluids. In *SIGGRAPH '99*, pages 121–128. ACM, 1999.

[17] K. Stephan, S. Klink, and C. Schraff. Assimilation of radar-derived rain rates into the convective-scale model cosmo-de at dwd. *Quarterly Journal of the Royal Meteorological Society*, 134(634):1315–1326, 2008.

[18] J. Tompson, K. Schlachter, P. Sprechmann, and K. Perlin. Accelerating eulerian fluid simulation with convolutional networks. In *Proceedings of Machine Learning Research*, pages 3424–3433, 2017.

[19] J. Xi, P. Lamata, W. Shi, S. Niederer, S. Land, D. Rueckert, S. G. Duckett, A. K. Shetty, C. A. Rinaldi, R. Razavi, et al. An automatic data assimilation framework for patient-specific myocardial mechanical parameter estimation. In *International Conference on Functional Imaging and Modeling of the Heart*, pages 392–400. Springer, 2011.

[20] J. Zehnder, R. Narain, and B. Thomaszewski. An advection-reflection solver for detail-preserving fluid simulation. *ACM Trans. Graph.*, 37(4):85:1–85:8, July 2018.