[Reviews · NeurIPS 2020]

Review 1

Summary and Contributions: The paper proposes to correct errors produced by PDE-solvers by inserting a neural network to the outputs of the (differentiable) PDE solver after each integration step and optimize the parameters of the neural network using backpropagation (through time) to compute the gradients wrt the neural network parameters. The experiments show that the proposed method work well.

Strengths: The idea is simple and the experimental results are good.

Weaknesses: The idea seems straightforward and it feels like it may have been tried before. After doing some search, I was not able to find this way of using neural networks in the literature. So, the idea seems novel.

Correctness: Mostly correct. Although I think that the paper could motivate better the problem being solved. What would be a real use case for the considered problem statement?

Clarity: Mostly well written. Although I did not understand completely the PRE approach. Does one use prior knowledge in designing the PRE-corrections? What is trained in the PRE-approach?

Relation to Prior Work: Yes

Reproducibility: Yes

Additional Feedback: - Is there benefit in using the differential PDE solver? It might be interesting to compare the proposed approach with just training a neural network to solve a PDE (without the differential PDE solver). - Line 52: What do you mean by explicit and implicit solvers? - Do steps of a differentiable simulator correspond to time steps? For example in Figure 1, t=200 corresponds to the 200-th step of the solver? The text says that the number of steps is up to 128, as shown in Fig. 1. - Line 177: What is "look-ahead trajectory per iteration"? What kind of iteration do you mean? - Line 202: It would be helpful to provide more details on the "constrained least-squares corrector". - If I understood the idea correctly, the computational graph of the model is a recurrent model with n consecutive blocks, each block is a combination of a differentiable solver and a conv net. The conv nets have shared parameters in each iteration. Is training of this model difficult because of the vanishing and exploding problems? - It would be helpful to draw the computational graph. - Paragraph starting on line 269: Do I understand correctly that in this experiment steps do not correspond to time but to iterations of a solver? It can be helpful to emphasise this more (maybe even in the introduction).


Review 2

Summary and Contributions: The authors propose to learn neural networks to correct (i.e., to reduce discretization errors of) PDE solutions. Their idea is to use differentiable PDE solves during training so that the correction neural network can be trained in an end-to-end manner, that is, both the correction step and the solution update step (by the solver) are taken into account in training. They provide intensive experimental results on several use cases.

Strengths: The technical motivation is clear, the method is reasonable, and the intensive experiments are convincing. The proposed method is a decent application of differentiable solvers and will be useful for applications where we need to solve PDEs for many settings.

Weaknesses: The computational burden can be discussed more. The main motivation for correcting coarse solutions is to achieve a good fidelity with fewer computation resources. So, readers will be very interested in how it can really reduce the computational burden in solving PDEs. The discussion on the runtime, which is detailed in the appendix, can be a part of the main text. Moreover, such an analysis should be provided for every type of experiment conducted.

Correctness: The experiments are convincing as to the types of problems tackled there.

Clarity: The statement of the paper is basically clear, but some points in the main text can be improved; some important details are deferred to the appendix. - What kinds of differentiable PDE solvers were adopted in the experiments? - How was the test datasets were created? Now it only says "test data sets whose parameter distributions differ from the ones of the training data set," but it should be elaborated as it is important to assess the adequacy of the experiments.

Relation to Prior Work: The related work part is too diverged. It should first focus on precisely *previous* studies, i.e., ones working on solution correction. Aren't there any researches that propose methods to correct coarse solutions of PDEs (or ODEs), not necessarily using neural networks? This lack of discussion makes the paper less convincing in terms of the novelty and the significance of the proposed method. If there are no previous studies that are directly comparable to the current one, it should be claimed so.

Reproducibility: Yes

Additional Feedback: Line 33-34: "We show that neural networks can only achieve optimal performance if they take the reaction of the solver into account." I think it is an overclaiming. We cannot tell if it's "can only" nor "optimal" from the experiments. Really minor points below. Line 130: typo, T s_t --> T r_t ? The broad impact section. The NeurIPS template says "Use unnumbered first level headings for this section, which should go at the end of the paper." ----- [After rebuttal] Thank you for the rebuttal. It is basically understandable, but I leave my score unchanged to be 6 because I cannot judge whether the main potential drawback (i.e., the lack of detailed discussion on computational burden) would be satisfactory covered in a revised version.


Review 3

Summary and Contributions: The paper propose a method to learn a correction function which is formulated as a neural network to reduce the numerical error of PDE solvers. The main contribution is that integrate the solver into the training process and thereby allow the correction function to interact with the PDE solver during training.

Strengths: The empirical evaluation is impressing. Several complex PDEs are tested to show the performance of the proposed error correction method.

Weaknesses: Not any theoretical analysis for the proposed method. In the proposed method, after training, the correction is a function of the numerical solution at current time, this may not be true. In principle, error should depend on the whole trajectory

Correctness: The empirical methodology of this paper is self-contained under the assumption that the correction is a function of current solution. However, I doubt this assumption may not be true.

Clarity: The paper is very hard to read. The idea of the paper is easy to understand, but too many conceptual interpretations make it difficult to get the main idea. The paper could be significantly simplified by introducing proper mathematical formula.

Relation to Prior Work: The difference from previous works is properly discussed.

Reproducibility: No

Additional Feedback: After rebuttal, the whole picture is more clear for me. But the details are still confusing. I am not sure if my question can be well addressed in the revision. I change the score to 5.


Review 4

Summary and Contributions: Problem: This paper focuses on PDE solver correction. It considers a fine grained simulation as reference and a coarse grained simulation as source. The goal is to learn a correction to the source with which the source simulation matches the reference. I can see a lot of meaningful applications such as weather forecast, where the reference simulation is hard to get. Technics: The authors point out the fact that the corrections would effect the PDE trajectory, therefore the training data one would collect under "non interacting" setting is not suitable for inference. In order to resolve this problem, the authors propose a solver-in-the-loop schema, which backpropagates gradients through the PDE solver on the source manifold to account for the influence of the corrector itself. Summary: This is a very well motivated work; the experiments are very well conducted; the high level idea are well explained and the logics are well organized.

Strengths: Topic: This work identifies an important practical problem (corrections themselves change PDE trajectory) and proposed an effective solution to it. Evaluation: The authors conduct a series of systemic experiments on multiple types of PDEs, and almost all gives desired results. Significance: The technics proposed here are especially suitable for PDE simulations where the reference can be observed yet very hard to simulate accurately (such as weather forecasting).

Weaknesses: Writing: It is hard to understand the notation and math without looking at the appendix. There are discrepancies between the main text and the appendix, e.g. line 132 in the main text defines the correction operator output as (original input + correction), while in the appendix, the correction operator output only contains the correction (line 25). Experiment Cost: It would be interesting to compare the cost of training the corrector verses directly running simulation on the reference manifold. Even though the later might be cheaper.

Correctness: The claims, methods, and empirical methodology seems correct to me.

Clarity: This paper is well written, although there seems to be minor inconsistencies in terms of notation.

Relation to Prior Work: Yes

Reproducibility: Yes

Additional Feedback: Questions: 1) In line 25 of the appendix, the correction is defined as a function of the current state s, regardless of the number of time steps since the initialization (line 21). Can you explain why? Intuitively, the longer the simulation on the source manifold, the larger the deviation from the reference would be, no?

[Author Response · NeurIPS 2020]

We appreciate the reviewer's valuable comments, and we were glad to read the positive comments regarding the technical motivation, idea, and our results. We also appreciate the thorough feedback for further improvements. We will address those issues in a future revision of our work.

**Review 1**: *What would be a real use case?* We believe our work can be applied to large variety of PDE simulations where the reference can be computed, but is costly to obtain. A particularly interesting application would weather prediction, where a simple differentiable solver could be augmented with a learned correction function to recover the costly predictions of operational forecasting systems.

*What is trained in the PRE-approach?* The prior knowledge used for PRE models is usually problem-dependent and makes use of a reduced version of the full PDE formulation. For example, the PRE model for the Navier-Stokes equations makes use of a divergence-free constraint. For details of the *constrained least-squares method* used for this model, we refer to Appendix A.2.

*Is there benefit in using the differentiable PDE solver?* It would be interesting to evaluate learned surrogate models that replace the source PDE in our training setup, as neural networks can provide gradients by construction. However, any errors introduced by the surrogate could yield sub-optimal gradient information, in turn deteriorating the quality of the learned correction.

*Do steps of a differentiable simulator correspond to time steps?* Yes, in our text "step" typically means time step. We use a normalized $\Delta t = 1$, so in Figure 1, $t$ directly indicates the number of recurrent time steps that were calculated to obtain the result shown. It's a good idea to add a visual overview of the *recurrent blocks*. Note that for the *CG solver* example, the steps correspond to the iteration of the CG solver.

For the *"look-ahead trajectory per iteration"*, the iteration denotes a single step of training. At each iteration, the weights of our model receive gradients from all look-ahead steps of the solver.

**Review 2**: *The computational burden can be discussed more.* We would be happy to include measurements for the other cases and discuss them in the main text. For example, the 3D example is particularly interesting but currently only mentioned at the end of the appendix. In this case, the regular reference solver needs ca. 957 seconds, compared to 12.5 seconds for a simulation with $\text{SOL}_{16}$.

*How were the test datasets created?* We chose offset parameters w.r.t. training data set or shifted distributions for initial conditions. Details of the test parametrizations are given towards the end of each first paragraph of the B.n sections in the appendix.

Our *PDE solvers* cover a variety of advection-diffusion problems. The B.n sections of the appendix also give details of the numerical methods we have implemented in a differentiable manner in our TensorFlow framework. We will also revise our text regarding taking solver reactions into account and clarifying novelty w.r.t. previous work.

**Review 3**: *In principle, error should depend on the whole trajectory.* While the error certainly accumulates and typically grows over the course of a full trajectory, our key hypothesis here is that a learned approach can nonetheless identify and correct a large part of the error function based on information from a single phase-space input. We do not claim that our method is able to perfectly correct the full error in each step, but our results demonstrate that a very significant portion is learnable. Moreover, our tests with models using history information consistently did not yield significance improvements. We are confident that this topic could be clarified easily in a revision.

A theoretical analysis for the highly non-linear cases we are targeting would be a interesting topic for future work, and we hope our work will inspire further research in this direction. As these are the only negative points mentioned in the review, we were surprised about the negative final assessment.

**Review 4**: Thank you for pointing out the *inconsistencies between main text and appendix*. We will correct this.

*It would be interesting to compare the cost of training versus a reference simulation.* Training a corrector is potentially costly. Training complexity primarily scales with the cost for the differentiable solver and the number of look-ahead steps. The complex SOL models can take more than a day of training time. However, we anticipate that the training cost will in practice quickly amortize as our models generalize well and can be re-used for a large number of new simulation runs. We will add training and reference simulation timings to Table 6 in the appendix.

*Why is the correction only defined as a function of the current state $\mathbf{s}_t$?* For the PDEs we consider, a single state actually uniquely describes its future evolution. We have experimented with additionally providing varying numbers of previous states $\mathbf{s}_{t-k}, ..., \mathbf{s}_{t-1}$ as input to our model, but our tests have not shown improvements. The tests indicate that the components of the error function that are learned with our approach can be reliably inferred from a single state $\mathbf{s}_t$. We can include these additional experiments to illustrate that providing additional states has a negligible influence on the learned corrector.

[Meta-Review · NeurIPS 2020]

Many reviewers felt strongly that this work represents an exciting research direction, in particular that it can be directly and effectively applied for dynamics where the reference can only be observed instead of accurately simulated. In your final revision, please address all changes promised in the rebuttal. Additional discussion of computational burden and avenues for future research should be added as well.